# Model Fusion via Retrofitting

**Phoomraphee Luenam** [* 1]  **Andreas Spanopoulos** [* 1]
**Amit Sant** [1]  **Sotiris Anagnostidis** [1]  **Thomas Hofmann** [1]  **Sidak Pal Singh** [1]

## Abstract

Model fusion seeks to combine independently trained neural networks into a single model without retraining, but is complicated by representational divergence arising from permutation invariance, random initialization, and heterogeneous training data. Existing methods struggle particularly in zero-shot settings under non-IID data distributions, and are often limited to specific architectures or pairwise fusion. We introduce a neuron-centric family of fusion algorithms that frames fusion as a principled representation-matching problem: intermediate neurons across parent models are grouped into target representations, which the fused model's corresponding sub-networks are then trained to approximate. Unlike prior work, our approach incorporates neuron attribution scores to bias alignment toward salient features, and can be applied to any architecture modularizable as a DAG of levels—empirically validated on VGGs, ResNets, and ViTs. Experiments across standard benchmarks show consistent improvements over existing fusion methods, with the largest gains in zero-shot and non-IID scenarios. Code is available at https://github.com/AndrewSpano/model-fusion-via-retrofitting.

## 1. Introduction

As modern Deep Neural Networks (DNNs) continue to grow in scale, retraining them on new data is often prohibitively expensive or infeasible, particularly in settings where data privacy must be preserved. Model fusion offers an appealing alternative: instead of retraining, one may combine *independently trained* models directly. Two influential contributions in this area are OTFusion (Singh & Jaggi, 2020) and Git

Re-Basin (Ainsworth et al., 2023). This line of work has been motivated in part by the Linear Mode Connectivity (LMC) conjecture, which posits that independently trained networks can be connected by low-loss paths (Frankle et al., 2020). While the empirical evidence for LMC is strong (Theus et al., 2026), prior studies reveal that even models trained on identical data may learn divergent internal features (Li et al., 2015), undermining the premise of weight averaging. While Ainsworth et al. (2023) argue that barriers vanish in sufficiently large networks, their own experiments highlight numerous failure modes, including simple architectures on small datasets, such as MLPs on MNIST when trained with SGD and a low learning rate.

In this work, we identify and address three key gaps in prior research on model fusion. **Reproducibility.** Many open-source implementations re-implement the same algorithm separately for each architecture, limiting generality and making systematic benchmarking difficult. **Base model quality.** Prior studies often report results using base models with accuracies below what is typically achieved by the same architectures. While the aim of those works was not to train state-of-the-art baselines, this raises the question of whether fusion methods could show similar improvements if the base models themselves were trained more thoroughly—even with simple techniques such as using stronger data augmentation (Zhang et al., 2018; Yun et al., 2019). **Heterogeneous data.** Experiments on models trained with different data distribution settings remain limited, with some works focusing on surrogate metrics such as loss or calibration rather than accuracy. In Section 5, we show empirically that existing methods struggle in zero-shot fusion under such heterogeneous conditions, which typically arise in the context of Federated Learning (FL).

**Contributions.** Motivated by these shortcomings, we introduce a family of neuron-centric fusion algorithms with the following key innovations: **(a) Casting fusion as a principled representation-matching problem**, yielding a two-stage algorithm that performs well across various data settings. **(b) Incorporating neuron saliency into alignment**, improving performance across our methods and enhancing existing approaches such as Git Re-Basin. **(c) A flexible open-source reimplementation of existing algorithms**, to facilitate systematic benchmarking in future work.

*Equal contribution [1]ETH Zürich. Correspondence to: Phoomraphee Luenam <luenamp@ethz.ch>, Andreas Spanopoulos <aspanopoulos@ethz.ch>.

*Proceedings of the 43rd International Conference on Machine Learning*, Seoul, South Korea. PMLR 306, 2026. Copyright 2026 by the author(s).

*Table 1.* Algorithm comparison in the context of model fusion.

| Algorithm | Linear layers | Trans-formers | Any modularizable architecture[a] | > 2 models | Different widths | Different depths | Benefits from imp. scores | High zero-shot acc. |
|---|---|---|---|---|---|---|---|---|
| OTFusion | ✓ | ✓ | ✗ | ✓ | ✓ | ✗ | ✗ | ✗ |
| Git Re-Basin[b] | ✓ | ✗ | ✗ | ✗ | ✗ | ✗ | ✓ | ✗ |
| ZipIt![c] | ✓ | ✗ | ✗ | ✓ | ✗ | ✗ | ✗ | ✗ |
| HF (Ours) | ✓ | ✓ | ✓ | ✗ | ✗ | ✓ | ✓ | ✓ |
| KF (Ours) | ✓ | ✓ | ✓ | ✓ | ✓ | ✓ | ✓ | ✓ |

[a] Theoretically applicable to architectures modularizable as a DAG of levels, empirically validated on VGGs, ResNets, and ViTs.
[b] Activations-based Git-Rebasin
[c] ZipIt! when fusing all layers

## 2. Related Work

### 2.1. Fusion Algorithms

**OTFusion** (Singh & Jaggi, 2020) formulates neuron alignment as a discrete optimal transport (OT) problem. Given multiple models, OTFusion selects an initial reference model, aligns each of the remaining models' layers to its layers via optimal transport on neuron activations, and averages the aligned parameters to produce a fused model. One limitation of OTFusion is that it was initially designed to handle only linear layers. Later work (Imfeld et al., 2024) adapted OTFusion to the transformer (Vaswani et al., 2017) architecture. However, it does not work out of the box with any architecture.

**Git Re-Basin** (Ainsworth et al., 2023) proposes three strategies to perform neuron alignment. In this work, we focus exclusively on the "Matching Activations" approach, which is most directly comparable to our methods. Activation-based Git Re-Basin finds a permutation matrix that minimizes the L2 distance between neuron activations across models, making it equivalent to OTFusion when the latter uses an activation-based ground metric. While effective, activation-based Git Re-Basin is limited to pairwise fusion, restricts itself to a one-to-one matching paradigm, and does not account for neuron saliency. We discuss our extension to incorporate importance scores in Appendix C. Subsequent work (Jordan et al., 2023) investigates the factors contributing to Git Re-Basin's limited zero-shot accuracy and proposes a remedy based on rescaling the weights of the fused model.

**ZipIt!** (Stoica et al., 2024) was proposed specifically to handle fusing base models trained on different tasks into a single multitask model without requiring retraining. Unlike Git Re-Basin and OTFusion, ZipIt! allows features from the same base model to be fused with one another, in addition to fusing features across base models, and does so greedily by iteratively combining the most highly correlated features.

Notably, ZipIt! allows the user to specify how many layers to "zip", leaving later layers unzipped and potentially retaining multiple classification heads in the fused result. In this work, we focus on the fully merged setting where all layers, including the classification heads, are fused, for consistency with prior model merging methods that produce a single unified model.

**Federated Learning algorithms**. In federated learning (FL), decentralized clients train local models on private data and periodically send updates to a central server, which aggregates them into a global model. Model fusion on non-IID data distributions is therefore central to FL. However, FL methods typically assume an iterative optimization loop in which the aggregated model is repeatedly redistributed to clients and further trained on private data. In contrast, one-shot (or zero-shot) model merging studies the fusion of *independently trained models without access to client data or additional training rounds*.

Among the most well-known methods in this domain are **Federated Averaging (FedAvg)** (McMahan et al., 2017) and **Federated Matching Averaging (FedMA)** (Wang et al., 2020). The former averages model weights across clients proportionally to the number of local updates or data samples. While simple and widely used, FedAvg performs poorly on independently trained models, as weights can diverge substantially in the presence of heterogeneous data, particularly for deeper architectures. FedMA aims to address this challenge by aligning neurons before averaging. However, it *requires redistributing and retraining* the fused model (using private client data) after aligning *every layer* to recover performance, making it a *non-zero-shot* fusion algorithm.

Finally, we briefly mention traditional approaches to aggregating knowledge from multiple models. **Ensembles**, which average the predictions of base models, typically represent an upper bound on the performance achievable by zero-shot

fusion, albeit *at the cost of additional computational overhead*. **Vanilla Averaging** blindly averages the weights of two identical models without alignment. In **Knowledge Distillation (KD)** (Hinton et al., 2015), a model is trained to predict soft targets produced by another model. While KD was initially developed for model compression, later work extended it to the multi-teacher setting (Asif et al., 2020; Lin et al., 2020).

### 2.2. Neuron Attribution

A novel aspect of our work is the optional incorporation of neuron attribution scores—commonly referred to as *neuron importance scores*—into the fusion process, in order to bias the preservation of salient features. As a naive baseline, **Uniform** importance assigns an equal weight of $\frac{1}{n}$ to each neuron in a layer of $n$ neurons. **Conductance** (Dhamdhere et al., 2019) extends *Integrated Gradients* (Sundararajan et al., 2017), which attributes feature importance by integrating gradients along a straight-line path from a baseline input to the actual input. Conductance applies the chain rule to propagate these importance scores to hidden neurons, enabling internal saliency estimation. **DeepLIFT** (Shrikumar et al., 2017) provides an alternative method for attributing importance scores to neurons. It computes each neuron's contribution by comparing its activation to a reference activation and propagating these differences through a modified chain rule. Unlike gradient-based methods, DeepLIFT can assign non-zero importance scores even when gradients are zero or poorly behaved, and requires only a single backward pass, making it computationally efficient.

## 3. Motivation

In a neural network, each layer of neurons can be interpreted as encoding a specific amount of information used by subsequent layers to produce an output (here, a "neuron" denotes a set of activations controlled by a common set of weights — this corresponds to a channel in a convolutional layer or a single embedding dimension in a transformer). Consequently, for the purpose of model fusion, a natural goal is to preserve the information contained in the neurons of the base models by ensuring that each base model neuron is closely represented by a neuron in the fused model. A natural metric for neuron closeness is the squared L2 (Euclidean) distance between the (pre)activations, which has already been used in the context of model fusion by Singh & Jaggi (2020) and Ainsworth et al. (2023). This motivates our definition of **representation cost** for a given level of the fused model. A natural extension to the raw squared L2 distance is to weigh the distances by *neuron importance scores* which intuitively penalizes the misrepresentation of more salient neurons.

We now introduce the notation. A deep neural network

(DNN) can be viewed as a function $f_{\mathbf{w}} : \mathbb{R}^d \mapsto \mathbb{R}^o$ parameterized by weights $\mathbf{w}$, where $d$ is the number of input features and $o$ is the number of output features. For many model architectures, $f_{\mathbf{w}}$ can be decomposed into $L$ sequential functions $f_{\mathbf{w}} = f_{\mathbf{w}_L}^L \circ \cdots \circ f_{\mathbf{w}_1}^1$ where $L$ denotes the depth of the model. These functions can be *arbitrarily* grouped into modules we call **levels**. For example, we can decompose $f_{\mathbf{w}} = f_{\mathbf{w}_3}^3 \circ f_{\mathbf{w}_2}^2 \circ f_{\mathbf{w}_1}^1$ into $f_{\mathbf{w}} = \widehat{f^2}_{\widehat{\mathbf{w}}_2} \circ \widehat{f^1}_{\widehat{\mathbf{w}}_1}$, where $\widehat{f^2}_{\widehat{\mathbf{w}}_2} = f_{\mathbf{w}_3}^3$ and $\widehat{f^1}_{\widehat{\mathbf{w}}_1} = f_{\mathbf{w}_2}^2 \circ f_{\mathbf{w}_1}^1$ are individual **levels**. Importantly, layers with branching (e.g. skip connections or transformer blocks) can be contained in a single level so that the functions may be composed sequentially as we demonstrate with ResNets and ViTs. The exact partitioning of the model into levels is a design choice, though modular structures such as classifier heads or transformer blocks make natural candidates.

For fusion, let $\mathcal{M} = \{M_1, M_2, \ldots, M_n\}$ be a collection of pretrained base models, each partitioned to have $L$ levels. To keep notation simple, we will define the **representation cost** for a fixed level $l$, assuming the fused model's weights at all previous levels are fixed. Let $\mathbf{z}^{M_k}$ be the output vector of model $M_k$ at this level and let $\mathbf{z} = \text{concat}(\mathbf{z}^{M_1}, \ldots, \mathbf{z}^{M_n}) \in \mathbb{R}^{d^{\mathcal{M}}}$ be the concatenated outputs, of total size $d^{\mathcal{M}}$. For a fused model $\mathcal{F}$ with weights $\mathbf{w}$ at level $l$, we write $\mathbf{z}^{\mathcal{F}_{\mathbf{w}}} \in \mathbb{R}^{d^{\mathcal{F}}}$ for its outputs with size $d^{\mathcal{F}}$. These outputs are typically the **activations** or **preactivations** at a given level. We denote by $s_j$ the importance score of neuron $j$ in the concatenated outputs $\mathbf{z}$. The **representation cost** of using weights $\mathbf{w}$ at level $l$ of the fused model $\mathcal{F}$ (for a given input $\mathbf{x}$) is then

$$J_{\mathbf{w}}(\mathbf{x}) = \sum_{j=1}^{d^{\mathcal{M}}} s_j \left( \min_{k \in \{1, \ldots, d^{\mathcal{F}}\}} \left\{ \left( z_k^{\mathcal{F}_{\mathbf{w}}}(\mathbf{x}) - z_j(\mathbf{x}) \right)^2 \right\} \right) \tag{1}$$

Intuitively, for each neuron in the concatenated base outputs $\mathbf{z}(\mathbf{x})$, we compute the squared L2 distance to **the closest neuron in the fused model output** $\mathbf{z}^{\mathcal{F}_{\mathbf{w}}}(\mathbf{x})$, which can be viewed as its **representative neuron**, and sum the resulting distances weighed by importance. To solve for the desired weights $\mathbf{w}$ of the fused model $\mathcal{F}$ at level $l$, we would, in principle choose $\mathbf{w}$ to minimize this cost.

Popular layer-wise fusion algorithms (Singh & Jaggi, 2020; Ainsworth et al., 2023) similarly optimize an L2 distance to obtain soft or hard permutation matrices for neuron alignment, then average the aligned weights of the base models layer by layer. In Section 5 we show empirically that these algorithms: **a)** fail to match base model performance in zero-shot fusion, i.e. they require a fine-tuning phase; and **b)** fail to achieve meaningful knowledge transfer in heterogeneous data regimes.

We hypothesize that these shortcomings arise for two reasons. **a)** Current algorithms treat each level in isolation,

tracking only past permutation or alignment matrices without accounting for how the fused model's intermediate representations evolve as the algorithm progresses through the levels. **b**) Neurons are averaged with equal importance, despite contributing unequally to a model's predictions on average. This is particularly problematic in heterogeneous data settings, where activations on unseen data may be noisy or irrelevant, and where different models may learn to extract fundamentally different features.

## 4. Proposed Method

To optimize the objective in Equation (1), we decouple it by introducing an auxiliary vector $\mathbf{T}$ of size $d^{\mathcal{F}}$, which we refer to as the **target vector**. This enables a more tractable decomposition of the cost function, yielding the following theorem.

**Theorem 1.** *Let $\mathbf{T} \in \mathbb{R}^{d^{\mathcal{F}}}$ be a vector whose components are the importance-weighted means of clustered outputs. Then the representation cost decomposes as:*

$$
J_{\mathbf{w}}(\mathbf{x}) = \underbrace{\sum_{k=1}^{d^{\mathcal{F}}} \sum_{j \in R_k} s_j \left( z_k^{\mathcal{F}}(\mathbf{x}) - T_k \right)^2}_{\text{approximation error}}
$$
$$
+ \underbrace{\sum_{k=1}^{d^{\mathcal{F}}} \sum_{j \in R_k} s_j \left( T_k - z_j(\mathbf{x}) \right)^2}_{\text{grouping error}} \tag{2}
$$

*where $R_k$ is the set (or cluster) of base model neurons that fused model neuron $k$ represents in the minimum-cost assignment in Equation (1).*

*Proof.* See Appendix A. □

We observe that the sum of $J_{\mathbf{w}}(\mathbf{x})$ over a batch is subdifferentiable with respect to $\mathbf{w}$ and that this objective could in principle be optimized with subgradient descent in the same spirit as the weighted K-means objective (Bottou & Bengio, 1994). However, we leave this for future work.

The decomposition yields two interpretable components. The **grouping error** measures how well the original neurons cluster together – specifically, how far each output $z_j$ is from the importance-weighted cluster center $T_k$ it was assigned to. The **approximation error** quantifies how closely the fused model reproduces these cluster centers via its output neurons $\mathbf{z}$.

For a batch of $B$ inputs $\mathbf{x}$, minimizing the total grouping error by constructing an optimal $\mathbf{T}$ through an effective clustering of the outputs $\{z_j\}$ is a critical challenge. This corresponds to the K-means problem in $\mathbb{R}^B$ which is known

to be NP-hard in general (Aloise et al., 2009). Nonetheless, practical approximation algorithms such as Lloyd's algorithm (Lloyd, 1982) or local search-based methods (Kanungo et al., 2002) yield effective solutions in practice.

After determining $\mathbf{T}$ (and hence the clusters $R_k$), we *retrofit* the weights of a level by minimizing the approximation error, which is equivalent to a weighted mean squared error loss. One may either keep the weights of keep the weights of previous levels frozen and optimize only the current level or optimize the whole subnetwork. In this work, we adopt the former:

$$
\mathbf{w}^* = \arg\min_{\mathbf{w}} \quad \mathbb{E}_{\mathbf{x} \sim D} \left[ \sum_{k=1}^{d^{\mathcal{M}}} \sum_{j \in R_k} s_j \left( z_k(\mathbf{x}) - T_k(\mathbf{x}) \right)^2 \right] \tag{3}
$$

This decomposition offers a more interpretable and stable optimization target by isolating the challenges of clustering and function approximation, rather than attempting to solve them jointly.

### 4.1. Proposed Algorithm

Following the derivation in Theorem 1, we propose a two-step algorithm to find weights $\mathbf{w}$ that minimize Equation (2) in expectation. The algorithm constructs the fused model bottom-up, iterating through the levels of the base models and producing the corresponding level of the fused model at each step.

At each level, the algorithm computes a matching or clustering of the base models' neurons to minimize the grouping error, then uses this grouping to compute importance-weighted centroids, which we use as the targets. Since these targets depend only on the base models' neurons, they can in principle be computed before any of the fused model's weights are determined.

Finally, we minimize the approximation error to obtain the fused model's weights, using a mini-batch to approximate the expected loss in practice. An illustration of our algorithm is provided in Figure 1, with high-level pseudocode in Algorithm 1 and implementation details in Appendix F.

### 4.2. Minimizing the Grouping/Approximation Errors

#### 4.2.1. GROUPING ERROR

For the grouping error, we distinguish between two cases based on the architecture of the base models and the constraints imposed on the assignment. Further details can be found in Appendix B.1.

(a) *Equal-size models with level-wise one-to-one matching.* This induces a cost that can be minimized using the Hungarian matching algorithm (Kuhn, 1955), as discussed in Section 3. We refer to this case as **Hungarian Fusion (HF)**.

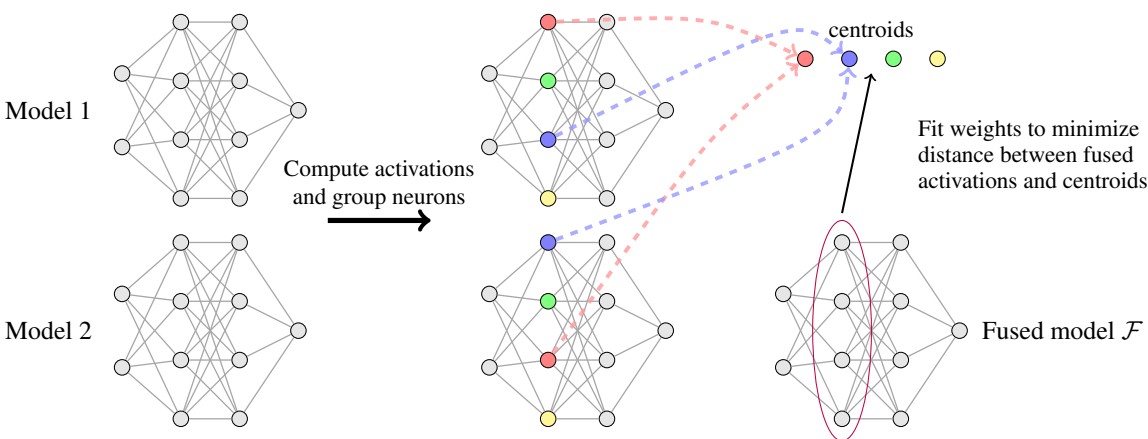

*Figure 1.* Overview of our method. Given two MLPs with two hidden layers of size 4, we compute the layer activations, cluster their neurons, and train the first layer of the fused model to match the centroids with the MSE loss. The process is repeated in subsequent levels.

---

**Algorithm 1** Model Fusion via Activation Retrofitting

**Require:** Base models $\mathcal{M} = \{M_k\}_{k=1}^n$, neuron importance scores $\{s_j^{M_k,l}\}_{k=1}^n{}_{l=1}^L$, fusion dataset $\mathbf{X} \in \mathbb{R}^{B \times d}$
**Ensure:** Fused model $\mathcal{F}$ with weights $W_{\mathcal{F}}$

1: **for** $l = 1, 2, \ldots, L$ **do**
2:    $\mathbf{z} \leftarrow \text{concat}\big(\mathbf{z}^{M_1,l}, \ldots, \mathbf{z}^{M_K,l}\big)$ {level-$l$ activations}
3:    $\mathbf{s} \leftarrow \text{concat}\big(\mathbf{s}^{M_1,l}, \ldots, \mathbf{s}^{M_K,l}\big)$ {level-$l$ imp. scores}
4:    Obtain clusters $(R_k)_{k=1}^{d^{\mathcal{F}}}$ for each output $z_j$
            {via hungarian matching or $k$-means}
5:    **for** $k = 1, \ldots, d^{\mathcal{F}}$ **do**
6:        $T_k \leftarrow \frac{\sum_{j \in R_k} s_j z_j}{\sum_{j \in R_k} s_j}$ {importance-weighted centroids}
7:    **end for**
8:    Define the objective
$$\mathcal{L}(\mathbf{w}) \leftarrow \sum_{m=1}^B \sum_{k=1}^{d^{\mathcal{F}}} \sum_{j \in R_k} s_j \big(z_k^{\mathcal{F}}(\mathbf{x}_m) - T_k(\mathbf{x}_m)\big)^2$$
9:    Optimize the weights $\mathbf{w}$ of level $l$ of $\mathcal{F}$
            $\mathbf{w} \leftarrow \arg\min_{\mathbf{w}} \mathcal{L}(\mathbf{w})$ {fit weights to centroids}
10: **end for**
11: **return** Fused model $\mathcal{F}$

---

(b) *General case with arbitrary model sizes.* The grouping problem becomes a general clustering task, which can be approximated using heuristic K-means algorithms as highlighted in Section 4.1. We refer to this general case as **K-means Fusion (KF)**.

### 4.2.2. APPROXIMATION ERROR

For the approximation error, we again distinguish two cases. Further details can be found in Appendix B.2.

(a) *Linear levels.* When all trainable levels are affine transformations (e.g. fully connected or convolutional layers), the

outputs $z_j$ are linear functions of the level weights $\mathbf{w}$, and the problem reduces to weighted least squares, which admits a closed-form solution. In this case, we project the base models' level outputs onto the image of the preceding fused level's outputs before running HF or KF. We refer to these algorithms as **HF-Linear** and **KF-Linear** respectively.

(b) *General case.* For arbitrary differentiable levels, we optimize Equation (3) via stochastic gradient descent (SGD). Note that for linear levels this objective is convex, reducing to case (a). We refer to the resulting algorithms as **HF-Gradient** and **KF-Gradient**.

### 4.2.3. GUARANTEES

We now present theoretical guarantees for Algorithm 1 under specific conditions.

**Theorem 2.** *Let the parametrized levels of the base models and fused model be affine functions, and let the fused model's weights for all levels preceding level $l$ be fixed. Then, compared with the decoupled objective in Equation (2) for level $l$:*

*(a) For two models with equal-sized levels and a one-to-one matching constraint,* ***Hungarian Fusion*** *returns an optimal solution.*

*(b) For an arbitrary number of models with possibly different numbers of neurons per level,* ***K-means Fusion*** *produces a solution whose representation cost is at most $(9 + \epsilon)$ times the optimal, when using the local-search algorithm of* Kanungo et al. (2002).

*Proof.* See Appendix A. □

Note that Theorem 2 assumes the weights of all levels preceding level $l$ to be fixed. This assumption is likely necessary in general, as jointly optimizing across levels is NP-hard even for networks with a single hidden layer of ReLU

activations (Goel et al., 2021).

# 5. Experiments

We evaluate our fusion algorithms across three distinct training regimes, each characterized by a different data distribution used to train the base models. This setup is designed to test the robustness and generality of our method. We benchmark our algorithms against baseline fusion algorithms, ensembles, vanilla averaging, KD, and Linear Probing (LP). Both KD and LP are special cases of our algorithm: the former treats the entire model as a single level, while the latter skips all layers except the classifier head.

**Base model performance in non-IID setups**  Before presenting our results, we highlight an important consideration when evaluating base model performance under non-IID conditions. In these settings, each model has access to only a small, often imbalanced portion of the dataset, which naturally limits its accuracy. For example, in 6-way splits, each model sees only 10–20% of the full data, leading to lower performance compared to centralized training. Despite these constraints, improvements in this setting are meaningful. Improving over weak, heterogeneous base models in a zero-shot setting is a challenging task, and our method demonstrates robustness where existing approaches fail.

**Experimental Settings**  We train base models with the **VGG** (VGG11), **ViT** (12 heads, 7 layers, and 384 hidden dimensions), and **ResNet** (18/34/50) model architectures. Unless otherwise indicated in the robustness studies, we use the same architecture for the base models and the fused model. We perform fusion using **Uniform**, **Conductance**, and **DeepLIFT** neuron importance scores. Scores were computed independently for each model using its training data. We provide further details in Appendices E and F.

## 5.1. Sharded Setup

We train base models on "sharded" data splits, which represent an extreme non-IID case where each model sees samples from a disjoint set of classes. This leads to class-specific overfitting and diverse learned representations across base models. This setup is typically considered in FL research, where it serves as a stress test for extremely skewed distributions. We further assume that the fusion dataset is skewed and drawn from one of the base models, mimicking the FL constraint that data cannot be shared across clients due to privacy and communication costs. If clients had access to the entire dataset, training directly on it would be more effective than model fusion. Details of the partitioning procedure are provided in Appendix D.

We evaluate all fusion algorithms in a *zero-shot* setting, where models are fused *without further retraining*. This

reflects the assumption that the fusion dataset is skewed, making further retraining unlikely to substantially improve performance. We report ViT results on CIFAR-100 in Table 2, with additional results on Tiny-ImageNet and VGGs for CIFAR-10 in Appendices G.1 and G.2.

Table 2 shows that our methods consistently outperform prior zero-shot baselines under extreme sharding. HF-Gradient achieves the best performance for the 2-way split (54.5%), outperforming KD (41.0%) and LP (45.3%). KF-Gradient remains robust as sharding increases, achieving 41.4% and 34.7% accuracy for 4-way and 6-way splits, outperforming OTFusion in all cases. By contrast, OTFusion collapses to near-random performance, highlighting the effectiveness of our approach in integrating models trained on disjoint label spaces without fine-tuning.

**BloodMNIST**  We present an experiment motivated by a potential real-world scenario on the BloodMNIST dataset (Yang et al., 2023). BloodMNIST contains 17,092 images of blood cells divided into 8 classes. 6 of the classes are white blood cells, while the remaining classes are erythroblasts and platelets. In this experiment, the dataset was sharded into one set containing the white blood cells and the other containing the erythroblasts and platelets. VGGs were trained on each set separately to distinguish the cell types within each set. *Fusion used only 400 samples from the white blood cell classes.* The results of fusing these models are shown in Table 3. HF-Linear and KF-Linear *outperform the ensemble* without requiring the sharing of private data while OTFusion and Git Re-Basin fail to outperform the base model that was given the white blood cell classes.

*Table 3.* **Test accuracy** comparison for VGGs fused on sharded splits of BloodMNIST. For each algorithm, we show the result with the importance scores that result in the best accuracy.

| Method | 2-WAY SPLIT |
|---|---|
| Individual Models | $76.1_{\pm0.1}, 22.8_{\pm0.0}$ |
| Ensemble | $88.4_{\pm3.0}$ |
| Vanilla Averaging | $15.2_{\pm7.1}$ |
| KD | $46.2_{\pm26.8}$ |
| LP | $66.3_{\pm15.4}$ |
| OTFusion | $22.9_{\pm6.2}$ |
| Git Re-Basin | $57.9_{\pm22.8}$ |
| HF-Linear (Ours) | $88.7_{\pm3.2}$ |
| KF-Linear (Ours) | $\mathbf{91.0}_{\pm2.7}$ |

**Comparison with ZipIt! on Imagenet-1k**  We compare KF-Gradient with ZipIt! on ImageNet-1k in Table 4. Following the experimental setup of Stoica et al. (2024), we partition ImageNet-1k into 5 disjoint subsets of 200 classes each. For each seed, we sample two partitions and fuse a model trained on the first with one trained on the second. We vary the number of fusion samples used by KF-Gradient

*Table 2.* **Test accuracy** comparison when fusing ViT networks on CIFAR-100 for **Sharded** splits. Full details in Table 17.

| Method | 2-WAY SPLIT | 4-WAY SPLIT | | 6-WAY SPLIT | | |
|---|---|---|---|---|---|---|
| Individual Models | $38.6_{\pm0.5}$ $37.2_{\pm0.6}$ | $20.4_{\pm0.2}$ $19.5_{\pm0.2}$ | $19.9_{\pm0.1}$ $19.2_{\pm0.3}$ | $14.3_{\pm0.2}$, $13.2_{\pm0.2}$ | $13.7_{\pm0.3}$ $12.8_{\pm0.3}$ | $13.5_{\pm0.2}$ $12.2_{\pm0.6}$ |
| Ensemble | $63.7_{\pm0.4}$ | $53.4_{\pm1.8}$ | | $45.4_{\pm2.0}$ | | |
| KD | $41.0_{\pm1.5}$ | $\mathbf{47.0}_{\pm0.8}$ | | $\mathbf{43.6}_{\pm0.8}$ | | |
| LP | $45.3_{\pm1.2}$ | $33.9_{\pm1.0}$ | | $27.2_{\pm1.0}$ | | |
| Transformer OTF acts | $2.3_{\pm0.5}$ | $1.2_{\pm0.2}$ | | $1.0_{\pm0.0}$ | | |
| Transformer OTF wts | $4.4_{\pm1.2}$ | $1.5_{\pm0.4}$ | | $1.3_{\pm0.4}$ | | |
| HF-Gradient (Ours) | $\mathbf{54.5}_{\pm1.2}$ | - | | - | | |
| KF-Gradient (Ours) | $54.4_{\pm1.1}$ | $41.4_{\pm1.0}$ | | $34.7_{\pm1.3}$ | | |

and find that KF-Gradient outperforms ZipIt! on both tasks even with fewer than half the samples, reflecting ZipIt!'s degraded performance when required to merge classifier heads rather than maintain separate ones.

*Table 4.* **Test accuracy** comparison for ResNet-50 models trained on two disjoint 200-category subsets (Task A and Task B) of ImageNet-1k, fused in the **Sharded** setup. KF-Gradient uses uniform importance scores. ZipIt! is configured to zip all layers. Results are averaged over 3 seeds.

| Method | JOINT | TASK A | TASK B |
|---|---|---|---|
| Model A | $41.1_{\pm0.1}$ | $82.3_{\pm0.2}$ | $0.0_{\pm0.0}$ |
| Model B | $41.1_{\pm0.6}$ | $0.0_{\pm0.0}$ | $82.1_{\pm1.1}$ |
| Ensemble | $65_{\pm1.1}$ | $64.4_{\pm1.5}$ | $65.5_{\pm3.4}$ |
| ZipIt! (25k samples) | $37.7_{\pm1.7}$ | $37.1_{\pm4.0}$ | $38.3_{\pm1.4}$ |
| KF-Gradient (1k samples) | $25.1_{\pm0.7}$ | $9.8_{\pm0.8}$ | $40.4_{\pm0.8}$ |
| KF-Gradient (2k samples) | $34.7_{\pm2.6}$ | $17.4_{\pm3.1}$ | $51.9_{\pm2.1}$ |
| KF-Gradient (5k samples) | $49.2_{\pm1.0}$ | $33.4_{\pm1.4}$ | $64.9_{\pm1.3}$ |
| KF-Gradient (10k samples) | $54.8_{\pm0.8}$ | $40.3_{\pm1.6}$ | $69.3_{\pm1.0}$ |
| KF-Gradient (25k samples) | $\mathbf{58.9}_{\pm1.0}$ | $\mathbf{46.1}_{\pm1.4}$ | $\mathbf{71.7}_{\pm0.9}$ |

### 5.2. Non-IID Setup

Similar to the Sharded setup, the data is split disjointly between models. In this case, however, multiple models may receive samples from the same class, but with skewed class distributions. We again evaluate all algorithms in *zero-shot* fusion. Experiments are conducted on VGG11 with CIFAR-10, with results averaged over five random seeds. Detailed results for the VGG-based experiments are provided in Appendix G.1.

### 5.3. Full Dataset Setup

Prior work on model fusion has predominantly considered the setting where models are trained on the full dataset, optionally followed by a fine-tuning phase to achieve performance gains over the individual base models. We include this setting in our evaluation for completeness. We merge ViT models on the CIFAR-100 and Tiny-ImageNet datasets,

as well as VGG models on CIFAR-10.

Results are reported in Tables 5, 6 and 15. In the zero-shot setting, Transformer OTFusion largely collapses for 2-way fusion, achieving near-random performance (4.3% on CIFAR-100 and 3.1% on Tiny-ImageNet), whereas KF-Gradient retains meaningful accuracy without retraining (63.0% on CIFAR-100 and 42.9% on Tiny-ImageNet). After fine-tuning, KF-Gradient consistently outperforms Transformer OTFusion (75.4% vs. 74.0% on CIFAR-100 and 54.2% vs. 53.8% on Tiny-ImageNet for 2-way fusion), substantially closing the gap with ensemble performance while maintaining single-model inference cost. These results demonstrate that KF-Gradient enables effective transformer fusion in regimes where prior approaches underperform. Further details on fine-tuning and pre-training are provided in Appendix F.3.

### 5.4. Robustness Studies

In this subsection, we present several experiments to probe the performance of our algorithms under various scenarios.

**Varying Fusion Dataset Size** Gradient-based fusion algorithms typically require substantial data, making them sensitive to the size of the available fusion set. This limitation can be mitigated through data augmentation. As shown in our ablation study on Table 7, augmenting a smaller fusion dataset (1k samples) substantially narrows the performance gap relative to using a larger dataset (5k samples).

*Table 7.* Zero-shot accuracy for VGG11 models trained on two non-IID CIFAR-10 splits. We vary the fusion dataset size for KF-Gradient. Base accuracies are 73.2 and 71.3. Data augmentation substantially reduces the gap between 5k and 1k samples.

| Fusion Dataset Size | Uniform | Conductance | DeepLIFT |
|---|---|---|---|
| 5K | $76.1_{\pm0.5}$ | $76.2_{\pm0.2}$ | $76.3_{\pm0.4}$ |
| 1K | $73.2_{\pm0.5}$ | $71.7_{\pm0.8}$ | $71.4_{\pm0.7}$ |
| 1K + Augmentations | $74.5_{\pm0.8}$ | $75.3_{\pm0.6}$ | $75.2_{\pm0.5}$ |

*Table 5.* **Test accuracy** comparison for ViTs trained and fine-tuned on CIFAR-100 in the **Full-Dataset** setup. Full details in Table 18.

| | Base Models | Ensemble | Zero-shot | | Fine-tuning | |
| --- | --- | --- | --- | --- | --- | --- |
| | | | Transformer OTFusion | KF-Gradient (Ours) | Transformer OTFusion | KF-Gradient (Ours) |
| 2-way fusion: | 73.9, 73.4 | $75.7_{\pm 0.3}$ +1.8 | $4.3_{\pm 0.2}$ -69.6 | $63.0_{\pm 1.2}$ -10.9 | $74.0_{\pm 0.4}$ +0.1 | $\mathbf{75.4}_{\pm 0.1}$ +1.5 |
| Inference Cost: | ×1 | ×2 | ×1 | ×1 | ×1 | ×1 |
| 4-way fusion: | 74.1, 73.6, 73.0, 72.9 | 76.6 +2.5 | 1.0 -73.1 | 57.5 -16.6 | 72.6 -1.5 | **75.6** +1.5 |
| Inference Cost: | ×1 | ×4 | ×1 | ×1 | ×1 | ×1 |

*Table 6.* **Test accuracy** comparison for ViTs trained and fine-tuned on Tiny-ImageNet in the **Full-Dataset** setup. Full details in Table 20.

| | Base Models | Ensemble | Zero-shot | | Fine-tuning | |
| --- | --- | --- | --- | --- | --- | --- |
| | | | Transformer OTFusion | KF-Gradient (Ours) | Transformer OTFusion | KF-Gradient (Ours) |
| | 52.7, 51.7 | $54.9_{\pm 0.4}$ +2.2 | $3.1_{\pm 0.2}$ -49.6 | $42.9_{\pm 0.3}$ -9.8 | $53.8_{\pm 0.1}$ +1.1 | $\mathbf{54.2}_{\pm 0.4}$ +1.5 |
| Inference Cost: | ×1 | ×2 | ×1 | ×1 | ×1 | ×1 |

**Fusing Models of Different Widths** The K-means variants of our algorithms are able to fuse models with different widths. In Table 16, we compare the performance of our algorithms with OTFusion, which also supports fusing layers with different width (while Git Re-Basin does not). Both KF-Gradient and KF-Linear are robust when fusing models of different widths, with KF-Gradient outperforming the base models and KF-Linear nearly matching them. In contrast, OTFusion struggles when fusing a VGG11 with another VGG11 with double or quadruple the width.

**ResNet Compression** Our fusion framework can be applied as a compression algorithm by *fusing a model into a smaller version of itself*. We compress a teacher trained on the full CIFAR-100 dataset into a smaller student using only 5000 samples. Compression is performed with KF-Gradient, relying solely on teacher activations to form target neurons, which can be interpreted as stage-wise knowledge distillation from a larger ResNet (He et al., 2016). We compare against KD using the same fusion dataset in Table 8. Despite the limited data, our method achieves meaningful knowledge transfer. Compressing ResNet-34 to ResNet-18 yields 70.6% accuracy, compared to below 50% for KD with random student initialization. For ResNet-50 to ResNet-18, our method achieves 59% accuracy while KD reaches only 44%. These results highlight that KD fails to transfer knowledge effectively when the available data is limited.

*Table 8.* Zero-shot accuracy for ResNet obtained by compressing a teacher model with KF-Gradient vs Knowledge Distillation. Teacher trained on full CIFAR-100, student has access to only 5k samples for compression.

| | | KF-Gradient | | | |
| --- | --- | --- | --- | --- | --- |
| Experiment | Teacher | Uniform | Conductance | DeepLIFT | KD |
| ResNet 34 → 18 | 82.4 | 70.2 | 70.5 | **70.6** | 45.9 |
| ResNet 50 → 18 | 83.2 | 57.8 | **59.0** | 57.6 | 44.0 |

**Fused Model Analysis and Insights** In Figure 2, we visualize the loss and accuracy landscapes of ViT base models trained on non-IID splits of CIFAR-10, alongside one of our fused models, using linear weight interpolation. The contour plots reveal flatter loss basins around the fused model, which is often indicative of improved generalization (Hochreiter & Schmidhuber, 1994).

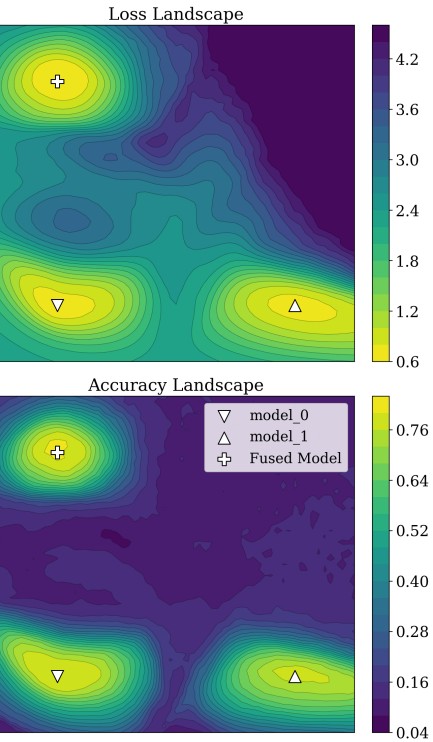

*Figure 2.* ViT Landscapes for CIFAR10, showing the fused model from KF-Gradient using DeepLIFT scores. Loss values were clipped to a maximum of 4.6 to keep a smooth gradient.

**Effect of Neuron Importance on Clustering** In Figure 3, we study how the neuron importance method affects the clustering stage of our algorithm. Recall that each neuron $j$ contributes to its cluster centroid as $T_k = \sum_{j \in R_k} s_j z_j / \sum_{j \in R_k} s_j$, so changing the importance scores $s_j$ alters both the centroids and the resulting assignments $R_k$. We take two VGG11 models trained on non-IID splits of CIFAR-10, fuse the first two layers with uniform scores using KF-Linear, and examine the importance-weighted K-means clustering at the third convolutional layer. The densest cluster (black) illustrates the effect clearly: under uniform importance, it remains concentrated in a narrow region, whereas Conductance and DeepLIFT pull the centroid toward high-importance neurons, broadening the cluster and changing which neurons are grouped together.

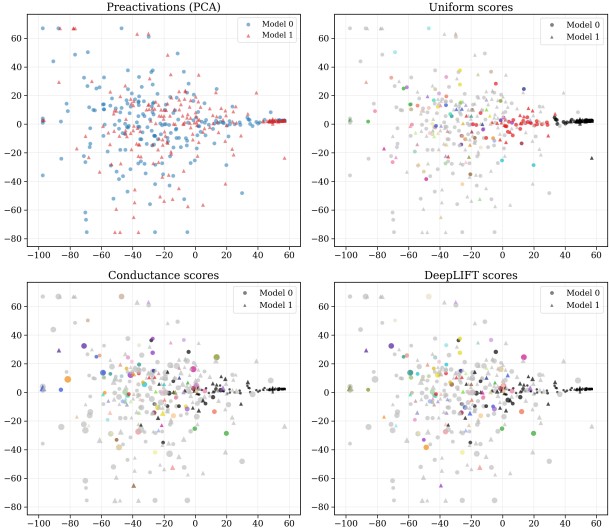

*Figure 3.* **Effect of neuron importance scores on clustering.** Preactivations and importance-weighted K-means clusterings of neurons from two VGG11 models trained on non-IID splits of CIFAR-10 at the third convolutional layer (256 channels per model, $K$=256). Neuron preactivations for 200 samples are projected to 2D via PCA. Black dots denote the densest cluster, gray dots denote singleton clusters, and other colors denote remaining clusters. Marker size encodes neuron importance relative to uniform ($1/n$). Under non-uniform attribution, the densest cluster shifts substantially; the adjusted Rand indices between the uniform assignments and those produced by Conductance and DeepLIFT are $0.74$ and $0.73$, respectively.

## 6. Limitations

Despite the strong performance of our proposed algorithms—often surpassing baselines and approaching ensemble performance—several areas for improvement remain. First, the gradient-based variants are sensitive to hyperparameters. In Appendix F.1, we identify a stable configuration that performs well across all experiments, though further automation of this selection remains an open direction.

Second, the performance of our gradient-based variants scales with the size of the fusion dataset. While more data improves results, this highlights a limitation in data efficiency; techniques such as data augmentation substantially mitigate this effect (Table 7), and recent work (Nasery et al., 2025) suggests that fusion using open-source data may enable data-free variants in the future.

Finally, our algorithms are slower than lightweight baselines such as OTFusion. As discussed in Appendix F.5, however, this overhead is offset in practice by eliminating the extensive fine-tuning that existing methods require to produce viable models.

## 7. Future Work

While our approach demonstrates strong empirical performance across a variety of fusion scenarios, several avenues remain open for further exploration.

**Experiments with LLMs.** With the growing prominence of large language models, it would be valuable to evaluate whether our algorithms yield improvements on such large-scale models, both in the context of fusion and compression.

**Automating hyperparameter selection and level partitioning.** Future work could explore principled methods for automatically tuning fusion hyperparameters, including the choice of level granularity and whether levels should end before or after activation functions.

**Alternative grouping methods.** Beyond K-means clustering and matching, other approaches to constructing layer-wise targets could be explored. One natural candidate is to form targets directly from the activations of the highest-importance neurons.

## 8. Conclusion

In this work, we introduced a novel neuron-aware approach to model fusion that supports fusing generic model architectures. Our algorithms, to our knowledge, are the first to successfully incorporate neuron importance scores in model fusion. Furthermore, our empirical results across diverse setups-including non-IID, sharded, and full-dataset regimes-consistently show that our fusion algorithms outperform existing baselines, particularly in the zero-shot scenario, and in some cases approach ensemble-level performance.

## Acknowledgments

Andreas Spanopoulos gratefully acknowledges financial support from the John S. Latsis Public Benefit Foundation, the Bodossaki Foundation, and the Union of Greek Shipowners.

## Impact Statement

This work concerns foundational research on model fusion algorithms. We do not foresee any negative applications beyond those broadly applicable to model fusion algorithms.

As with other fusion methods, negative societal impacts may follow from using biased or harmful models as a base model to perform fusion as the fused model may contain the biases / harmful potential of the base model.

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

# A. Proofs

We first present the proof of Theorem 1.

*Proof.* We decompose the cost of Equation (1) as follows, omitting the input $\mathbf{x}$ for notational clarity:

$$
\begin{aligned}
J_{\mathbf{w}} &= \sum_{j=1}^{d^{\mathcal{M}}} s_j \min_k \left\{ \left( z_k^{\mathcal{F}} - z_j \right)^2 \right\} \\
&= \sum_{j=1}^{d^{\mathcal{M}}} s_j \left( z_{k_j}^{\mathcal{F}} - z_j \right)^2 \quad \text{where } k_j = \arg\min_k \left( z_k^{\mathcal{F}} - z_j \right)^2 \implies j \in R_{k_j} \\
&= \sum_{j=1}^{d^{\mathcal{M}}} s_j \left[ \left( z_{k_j}^{\mathcal{F}} - T_{k_j} \right)^2 + 2 \left( z_{k_j}^{\mathcal{F}} - T_{k_j} \right) \left( T_{k_j} - z_j \right) + \left( T_{k_j} - z_j \right)^2 \right] \quad \left( \pm T_{k_j} \right)
\end{aligned}
\tag{4}
$$

When setting $k_j = \arg\min_k \left( z_k^{\mathcal{F}} - z_j \right)^2$, we break ties arbitrarily such that the sets $(R_k)_k^{d^{\mathcal{F}}}$ are non-overlapping and cover all base model neurons.

Since no constraints are imposed on the target vector $\mathbf{T}$, we retain the flexibility to define it in a way that simplifies the optimization. Specifically, if we rearrange the summation in Equation (4) into two nested sums — first over neurons in the fused model ($k = 1, \ldots, d^{\mathcal{F}}$), then over base model neurons $j$ assigned to each $k$ (i.e., $k_j = \arg\min_k \left( z_k^{\mathcal{F}} - z_j \right)^2$) and define $T_k$ as the importance-weighted mean of the assigned outputs, i.e., $T_k = \frac{\sum_{j \in R_k} s_j z_j}{\sum_{j \in R_k} s_j}$, then the cross-term in Equation (4) vanishes:

$$
\begin{aligned}
\sum_{j=1}^{d^{\mathcal{M}}} 2 s_j \left( z_{k_j}^{\mathcal{F}} - T_{k_j} \right) \left( T_{k_j} - z_j \right) &= 2 \sum_{k=1}^{d^{\mathcal{M}}} \sum_{j \in R_k} s_j \left( z_k^{\mathcal{F}} - T_k \right) \left( T_k - z_j \right) \\
&= 2 \sum_{k=1}^{d^{\mathcal{M}}} \left( z_k^{\mathcal{F}} - T_k \right) \sum_{j \in R_k} s_j \left( T_k - z_j \right) \\
&= 2 \sum_{k=1}^{d^{\mathcal{M}}} \left( z_k^{\mathcal{F}} - T_k \right) \sum_{j \in R_k} s_j \left( \frac{\sum_{i \in R_k} s_i z_i}{\sum_{i \in R_k} s_i} - z_j \right) \\
&= 2 \sum_{k=1}^{d^{\mathcal{M}}} \left( z_k^{\mathcal{F}} - T_k \right) \left( \sum_{i \in R_k} s_i z_i - \sum_{j \in R_k} s_j z_j \right) \\
&= 0
\end{aligned}
$$

Therefore, Equation (4) becomes:

$$
\begin{aligned}
J_{\mathbf{w}} &= \sum_{j=1}^{d^{\mathcal{M}}} s_j \left[ \left( z_{k_j}^{\mathcal{F}} - T_{k_j} \right)^2 + \left( T_{k_j} - z_j \right)^2 \right] \\
&= \sum_{k=1}^{d^{\mathcal{M}}} \sum_{j \in R_k} s_j \left[ \left( z_k^{\mathcal{F}} - T_k \right)^2 + s_j \left( T_k - z_j \right)^2 \right] \quad \text{(re-expressing the sum over output neurons)}
\end{aligned}
$$

$\square$

We now present the proof of Theorem 2.

*Proof.* We analyze Hungarian Fusion and K-means Fusion separately.

**(a) Optimality of Hungarian Fusion.** As established in Section 4.1, the decoupled objective Equation (2) separates into two terms: the *grouping error* and the *approximation error*. For linear levels, the outputs $z_k$ are affine functions of the weights $\mathbf{w}$, and thus the approximation error reduces to a weighted least squares problem which admits a closed-form solution.

Consequently, minimizing the total cost reduces to minimizing the grouping error. In the special case of two models with equal-sized layers and one-to-one neuron matching, this corresponds to a Linear Sum Assignment Problem (LSAP) with importance-weighted squared error as the cost matrix. The Hungarian algorithm solves this problem exactly in polynomial time (Kuhn, 1955), hence HF returns the optimal solution.

**(b) Approximation bound for K-means Fusion.** Consider a fixed assignment of base model neurons, where neuron $j$ is assigned to fused neuron $k_j$. The total representation cost associated with base neurons assigned to the $k$-th fused neuron at level $l$ is $\sum_{j \in R_k} s_j (z_k^{\mathcal{F}} - z_j)^2$ for a single sample. Stacking over samples gives $\sum_{j \in R_k} s_j \|\mathbf{z}_k^{\mathcal{F}} - \mathbf{z}_j\|^2$, where $\mathbf{z}_k^{\mathcal{F}}, \mathbf{z}_k \in \mathbb{R}^n$ are column vectors with entries corresponding to the preactivation for each input sample. Let the previous level's activations be $\mathbf{X} \in \mathbb{R}^{n \times d^{\mathcal{M}}}$. Since $z_k^{\mathcal{F}}$ is a linear function of $\mathbf{X}$, we have $\mathbf{z}_k^{\mathcal{F}} = \mathbf{X}\mathbf{w}_k$, where $\mathbf{w}_k$ are the weights of the $k$-th fused neuron (appending a column of ones to $\mathbf{X}$ if a bias term is present). Let $\mathbf{P} = \mathbf{X}(\mathbf{X}^T\mathbf{X})^+\mathbf{X}^T$ denote the projection matrix onto the column space of $\mathbf{X}$, so that $\mathbf{I} - \mathbf{P}$ projects onto its orthogonal complement, with $\mathbf{P}\mathbf{X} = \mathbf{X}$ and $(\mathbf{I} - \mathbf{P})\mathbf{X} = 0$. Then:

$$
\begin{aligned}
\sum_{j \in R_k} s_j \|\mathbf{z}_k^{\mathcal{F}} - \mathbf{z}_j\|^2 &= \sum_{j \in R_k} s_j \|\mathbf{X}\mathbf{w}_k - \mathbf{z}_j\|^2 \\
&= \sum_{j \in R_k} s_j \|\mathbf{P}(\mathbf{X}\mathbf{w}_k - \mathbf{z}_j) + (\mathbf{I} - \mathbf{P})(\mathbf{X}\mathbf{w}_k - \mathbf{z}_j)\|^2 \\
&= \sum_{j \in R_k} s_j \left( \|\mathbf{P}(\mathbf{X}\mathbf{w}_k - \mathbf{z}_j)\|^2 + \|(\mathbf{I} - \mathbf{P})(\mathbf{X}\mathbf{w}_k - \mathbf{z}_j)\|^2 \right) \quad \text{(orthogonality)} \\
&= \sum_{j \in R_k} s_j \|\mathbf{X}\mathbf{w}_k - \mathbf{P}\mathbf{z}_j\|^2 + \sum_{j \in R_k} s_j \|(\mathbf{I} - \mathbf{P})\mathbf{z}_j\|^2
\end{aligned}
$$

Letting $\bar{\mathbf{z}}_k = \frac{\sum_{j \in R_k} s_j \mathbf{z}_j}{\sum_{j \in R_k} s_j}$, so that $\sum_{j \in R_k} s_j (\mathbf{P}\bar{\mathbf{z}}_k - \mathbf{P}\mathbf{z}_j) = 0$, we expand the first term:

$$
\begin{aligned}
\sum_{j \in R_k} s_j \|\mathbf{X}\mathbf{w}_k - \mathbf{P}\mathbf{z}_j\|^2 &= \sum_{j \in R_k} s_j \|(\mathbf{X}\mathbf{w}_k - \mathbf{P}\bar{\mathbf{z}}_k) + (\mathbf{P}\bar{\mathbf{z}}_k - \mathbf{P}\mathbf{z}_j)\|^2 \\
&= \sum_{j \in R_k} s_j \|\mathbf{X}\mathbf{w}_k - \mathbf{P}\bar{\mathbf{z}}_k\|^2 + 2(\mathbf{X}\mathbf{w}_k - \mathbf{P}\bar{\mathbf{z}}_k)^T \underbrace{\sum_{j \in R_k} s_j (\mathbf{P}\bar{\mathbf{z}}_k - \mathbf{P}\mathbf{z}_j)}_{0} \\
&\quad + \sum_{j \in R_k} s_j \|\mathbf{P}\bar{\mathbf{z}}_k - \mathbf{P}\mathbf{z}_j\|^2 \\
&= \sum_{j \in R_k} s_j \|\mathbf{X}\mathbf{w}_k - \mathbf{P}\bar{\mathbf{z}}_k\|^2 + \sum_{j \in R_k} s_j \|\mathbf{P}\bar{\mathbf{z}}_k - \mathbf{P}\mathbf{z}_j\|^2
\end{aligned}
$$

Substituting back and summing over $k$ yields the full layer representation cost:

$$
\sum_{k=1}^{d^{\mathcal{F}}} \sum_{j \in R_k} s_j \|\mathbf{z}_k^{\mathcal{F}} - \mathbf{z}_j\|^2 = \sum_{k=1}^{d^{\mathcal{F}}} \sum_{j \in R_k} s_j \|\mathbf{X}\mathbf{w}_k - \mathbf{P}\bar{\mathbf{z}}_k\|^2 + \sum_{k=1}^{d^{\mathcal{F}}} \sum_{j \in R_k} s_j \|\mathbf{P}\bar{\mathbf{z}}_k - \mathbf{P}\mathbf{z}_j\|^2 + \sum_{j=1}^{d^{\mathcal{M}}} s_j \|(\mathbf{I} - \mathbf{P})\mathbf{z}_j\|^2
$$

The last term is independent of both the assignment $k_j$ and the weights $\mathbf{w}_k$, and is therefore always incurred.

Let $OPT$ denote the optimal representation cost. Since the first term is non-negative, any solution incurs cost at least $\sum_{k=1}^{d^{\mathcal{F}}} \sum_{j \in R_k} s_j \|\mathbf{P}\bar{\mathbf{z}}_k - \mathbf{P}\mathbf{z}_j\|^2 + \sum_{j=1}^{d^{\mathcal{M}}} s_j \|(\mathbf{I} - \mathbf{P})\mathbf{z}_j\|^2$. Letting $OPT_{\text{grouping}}$ denote the minimum of

$\sum_{k=1}^{d^{\mathcal{F}}} \sum_{j \in R_k} s_j \|\mathbf{P}\bar{\mathbf{z}}_k - \mathbf{P}\mathbf{z}_j\|^2$ over all assignments, we obtain:

$$OPT \geq OPT_{\text{grouping}} + \sum_{j=1}^{d^{\mathcal{M}}} s_j \|(\mathbf{I} - \mathbf{P})\mathbf{z}_j\|^2$$

We now consider KF. It first projects the activations onto the image of $\mathbf{X}$, then finds K-means clusters of $(\mathbf{P}\mathbf{z}_j)_{j=1}^{d^{\mathcal{M}}}$ to minimize $\sum_{k=1}^{d^{\mathcal{F}}} \sum_{j \in R_k} s_j \|\mathbf{P}\bar{\mathbf{z}}_k - \mathbf{P}\mathbf{z}_j\|^2$. It then fits $\mathbf{w}_k$ by solving a weighted least squares problem (see Appendix B.2), achieving $\|\mathbf{X}\mathbf{w}_k - \mathbf{P}\bar{\mathbf{z}}_k\|^2 = 0$, since $\mathbf{P}\bar{\mathbf{z}}_k$ lies in the column space of $\mathbf{X}$ by construction. The total representation cost for KF is therefore:

$$\sum_{k=1}^{d^{\mathcal{F}}} \sum_{j \in R_k} s_j \|\mathbf{P}\bar{\mathbf{z}}_k - \mathbf{P}\mathbf{z}_j\|^2 + \sum_{j=1}^{d^{\mathcal{M}}} s_j \|(\mathbf{I} - \mathbf{P})\mathbf{z}_j\|^2$$

where the first term is the weighted K-means cost over the projected activations and the second is unavoidable.

While K-means is NP-hard in general (Aloise et al., 2009), we may use the local-search algorithm of Kanungo et al. (2002), which provides a $(9 + \epsilon)$-approximation for the weighted K-means cost under squared Euclidean distance. This gives a clustering where $\sum_{k=1}^{d^{\mathcal{F}}} \sum_{j \in R_k} s_j \|\mathbf{P}\bar{\mathbf{z}}_k - \mathbf{P}\mathbf{z}_j\|^2 \leq (9 + \epsilon) OPT_{\text{grouping}}$. Therefore, the total cost attained by KF satisfies:

$$(9 + \epsilon) OPT_{\text{grouping}} + \sum_{j=1}^{d^{\mathcal{M}}} s_j \|(\mathbf{I} - \mathbf{P})\mathbf{z}_j\|^2 \leq (9 + \epsilon)\left(OPT_{\text{grouping}} + \sum_{j=1}^{d^{\mathcal{M}}} s_j \|(\mathbf{I} - \mathbf{P})\mathbf{z}_j\|^2\right) \leq (9 + \epsilon) OPT$$

establishing that KF is a $(9 + \epsilon)$-approximation for the layer representation cost.

$\square$

# B. Efficiently Minimizing the Fusion Errors

## B.1. Minimizing the Grouping Error

For the special case (a), we can re-express Equation (1) as a sum over the two base models:

$$J_{\mathbf{w}} = \sum_{j=1}^{d^{M_1}} s_j^{M_1} \min_k \left\{ \left(z_k^{\mathcal{F}} - z_j^{M_1}\right)^2 \right\} + \sum_{j=1}^{d^{M_2}} s_j^{M_2} \min_k \left\{ \left(z_k^{\mathcal{F}} - z_j^{M_2}\right)^2 \right\} \tag{5}$$

This cost admits a decomposition analogous to Equation (2). We define the cost of matching neuron $j_1$ of $M_1$ to neuron $j_2$ of $M_2$ as the cost of approximating the importance-weighted centroid $T_{j_1, j_2}$ of the resulting cluster by any neuron of the fused model $\mathcal{F}$; equivalently, one may use pairwise distances between level outputs directly as the cost matrix. The Hungarian algorithm (Kuhn, 1955) then finds a one-to-one matching minimizing Equation (5).

For the general case (b), we use Lloyd's algorithm (Lloyd, 1982) for K-means, which offers a favorable tradeoff between simplicity and effectiveness. K-means++ initialization typically yields improved clusterings. In this setting, neurons are treated as data points: the features of each neuron are the values it takes—e.g. its activations or preactivations—across the samples $\mathbf{x}$ in the dataset. Clustering is then performed over these vectors using importance-weighted K-means, with the number of clusters set to the desired number of neurons in the fused layer. Once clusters are formed, we compute importance-weighted centroids, yielding the target matrix $\mathbf{T} \in \mathbb{R}^{B \times d^{\mathcal{F}}}$, where $B$ is the batch size.

## B.2. Minimizing the Approximation Error

For the special case where a level is a linear function of its weights $\mathbf{w}$, i.e. $\mathbf{z} = \mathbf{X}\mathbf{w}$ for some $\mathbf{X}$, the approximation error in Equation (3) admits a closed-form weighted-MSE solution:

$$\mathbf{w}^* = \left(\mathbf{X}^\top \mathbf{S} \mathbf{X}\right)^+ \mathbf{X}^\top \mathbf{S} \mathbf{T}$$

where $\mathbf{S} = \text{diag}(s_1, \ldots, s_{d^{\mathcal{M}}})$ and $(\cdot)^+$ denotes the Moore-Penrose pseudoinverse.

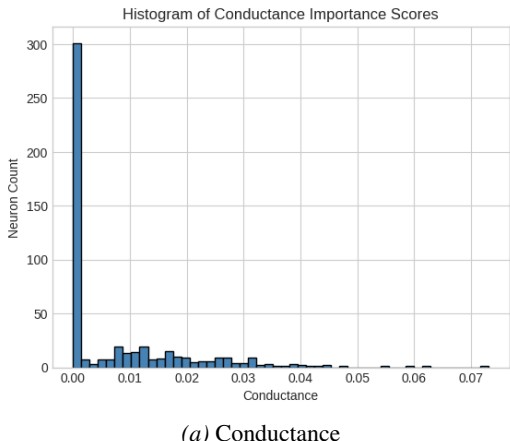
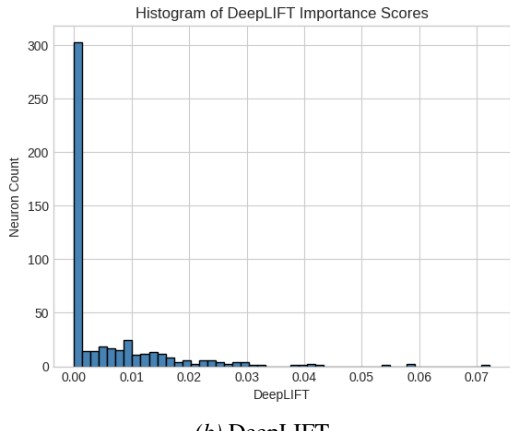

*(a)* Conductance        *(b)* DeepLIFT

*Figure 4.* Histogram of Conductance and DeepLIFT Importance Scores for the Last Convolutional Layer of a VGG11 on one non-IID split (split by two) on CIFAR-100.

For the general case, where a level is a non-linear differentiable function of its weights, a local minimum can be obtained via SGD. In practice, we use Adam (Kingma & Ba, 2015) or AdamW (Loshchilov & Hutter, 2017). We also opt for plain MSE rather than weighted MSE in the approximation step. With weighted MSE, neurons with low importance scores receive negligible gradient signal, which—while only marginally affecting the representation cost at the current level—can produce noisy or poorly structured intermediate representations in later levels. This issue is particularly pronounced in non-IID settings, where many neurons tend to receive importance scores close to zero, as illustrated in Figure 4.

## C. Comparison of Hungarian Fusion with Existing Algorithms

**Hungarian Fusion** is closely related to both **OTFusion** (Singh & Jaggi, 2020) and the activation-based variant of **Git Re-Basin** (Ainsworth et al., 2023). In the setting of equal-size models with level-wise one-to-one matching, all three methods construct a matching between neurons—equivalently, a transport map or permutation matrix aligning the second model to the first—by solving a minimum-cost matching problem.

A key distinction is that HF accounts for the effect of refitting previous levels, and correspondingly refits the weights of the current level to minimize accumulated error. As our experimental results show, this substantially improves zero-shot performance.

Regarding neuron importance scores, Singh & Jaggi (2020) proposed using importance as the probability measure assigned to each neuron in the optimal transport formulation, while Ainsworth et al. (2023) did not address importance scores in Git Re-Basin. In our experiments, we follow Singh & Jaggi (2020) for OTFusion; for Git Re-Basin, we weight each neuron's contribution during averaging by its importance score, in the same manner as HF.

## D. Data Partitioning Regimes

### D.1. Non-IID Splits

To simulate non-IID splits, we use a Dirichlet distribution to create imbalanced class distributions across models. Let $N_c$ denote the number of data points in class $c$, and $\alpha_k$ the concentration parameter for model $k$. The data for each class $c$ is distributed across models as:

$$\text{split}_k \sim \text{Dir}(\alpha_1, \ldots, \alpha_k)$$

where $\text{Dir}(\cdot)$ denotes the Dirichlet distribution. The concentration parameters $\alpha_k$ are arranged in an increasing sequence determined by the parameter `min_max_ratio`:

```
alpha_min = 1.0
min_max_ratio = 0.2
alpha_max = alpha_min / min_max_ratio
```

```
alphas = linspace(alpha_min, alpha_max, min_max_ratio)
```

A smaller ratio results in greater disparity between splits, amplifying heterogeneity. The Dirichlet-distributed probabilities determine the number of samples assigned to each model for class $c$, ensuring that splits exhibit non-uniform class distributions. Random shuffling of *class indices* and *concentration parameters* for each class $c$ introduces additional randomness in the resulting splits.

### D.2. Sharded Splits

In the sharded partitioning regime, each model receives examples from a disjoint subset of classes, so no two models share any classes in their local datasets. This simulates a strongly heterogeneous setting based on class exclusivity rather than distributional imbalance.

Let $\mathcal{C}$ denote the set of all classes in the dataset. The class set is first randomly permuted and then evenly partitioned into $K$ disjoint subsets, where $K$ is the number of models. Each subset $\mathcal{C}_k$ is assigned to model $k$, and all examples belonging to classes in $\mathcal{C}_k$ are included in that model's local dataset:

$$\bigcup_{k=1}^{K} \mathcal{C}_k = \mathcal{C}, \quad \mathcal{C}_i \cap \mathcal{C}_j = \emptyset \quad \forall i \neq j$$

## E. Model Training Details

We trained **VGG**, **ViT**, and **ResNet** models on NVIDIA RTX A5000 GPUs.

**VGG.** We use the VGG11 architecture, based on the implementation of Singh & Jaggi (2020).[1]

**ViT.** Our implementation is based on omihub777,[2] with the following model architecture settings:

| Hyperparameter | CIFAR-100 | Tiny-ImageNet |
|---|---|---|
| Patch Size | 4 | 8 |
| Encoder Blocks | 7 | 7 |
| Attention Heads | 12 | 12 |
| Hidden Dimension | 384 | 384 |
| MLP Hidden Dimension | 1536 | 384 |

**ResNet.** We experiment with ResNet18, ResNet34, and ResNet50, based on the implementation of Leclerc et al. (2022).

All base models are trained with the following hyperparameters:

| Hyperparameter | Value |
|---|---|
| Warmup Epochs | 5 |
| Minimum Learning Rate | $10^{-5}$ |
| Learning Rate | $10^{-3}$ |
| Label Smoothing | 0.1 |
| Batch Size | 128 |

The following PyTorch augmentations were applied during training: `RandomCrop`, `RandomHorizontalFlip`, `Normalize`, and additional augmentations following omihub777, based on AutoAugment (Cubuk et al., 2019). Base models were trained using a gradual warmup schedule combined with PyTorch's `CosineAnnealingLR` scheduler, until convergence; the exact number of epochs varies by model and dataset.

Full hyperparameter details and all code for reproduction are available in our open-source repository at `https://github.com/AndrewSpano/model-fusion-via-retrofitting`.

---

[1]`https://github.com/sidak/otfusion`
[2]`https://github.com/omihub777/ViT-CIFAR`

# F. Fusion Implementation Details

## F.1. Fusion Hyperparameters

Our proposed fusion algorithms include both linear and gradient-based variants, each with distinct hyperparameter considerations.

**Linear Variants.** The linear variants (HF-Linear and KF-Linear) require minimal hyperparameter tuning. The primary decisions concern whether to normalize (i) neuron outputs or (ii) neuron importance scores. We found that omitting normalization typically yielded better results across all data partitioning settings, likely because normalization can distort relative differences in neuron output magnitude that are informative for matching and clustering. The L2 regularization strength $\lambda$ used when computing the pseudoinverse is the other key hyperparameter; we found $\lambda = 10^{-3}$ for HF-Linear and $\lambda = 10$ for KF-Linear to work well across most experiments.

**Gradient-based Variants.** The gradient-based variants introduce a broader set of hyperparameters:

1. **Optimization:** learning rate, weight decay, number of gradient steps per level, batch size, validation split, and validation patience.

2. **Initialization:** initialization method used for the weights of the fused model at each level before running gradient descent.

3. **Clustering:** number of clusters (typically matched to the fused model's layer width), K-means++ initialization, early stopping criteria, and whether to normalize neuron outputs prior to clustering.

4. **Importance Weighting:** whether and how to incorporate neuron importance scores into clustering and loss weighting.

While this added complexity increases flexibility, it requires careful tuning for stable optimization. To mitigate this, we conducted extensive experiments and identified a **single** hyperparameter configuration that generalizes well across datasets (CIFAR-10, CIFAR-100), architectures (VGG11, ViT), and fusion regimes (Full Dataset, Non-IID, Sharded), reported in Table 9.

*Table 9.* Hyperparameters used for HF-Gradient and KF-Gradient

| Hyperparameter | Value |
|---|---|
| Optimizer | AdamW |
| Learning Rate | $10^{-3}$ |
| Epochs | 100 |
| Weight Decay | $10^{-4}$ |
| Perturbation $\epsilon$ | 1.0 |
| Normalize Activations | False |
| Train Batch Size | 32 |
| Val Split | 0.1 |
| Head Weights | False |

Here, "Head Weights" refers to weighting each model's final logits by the proportion of training samples seen per class. In practice this improves accuracy marginally, but at the cost of calibration—test loss increases, which may not be a desirable tradeoff in all settings.

## F.2. Model Partitioning Schemes

Due to the flexibility of our algorithm, we had the freedom to develop our own partition. In practice we as we primarily tested on like models, there were obvious answers that we used.

For VGG11s, each level contained only a single convolutional or linear layer to be aligned or an activation function, which did not need to be aligned.

For ViTs, each level corresponded to an encoder block except for the last one which corresponded to the classifier head.

### F.3. Post-Fusion Fine-tuning Hyperparameters

For full-dataset fusion, a fine-tuning phase is shown to improve fused model performance above that of the base models. We use the same optimizer as was used to train the corresponding base models, together with PyTorch's `CosineAnnealingWarmRestarts` scheduler. The hyperparameters for this phase are as follows:

| Hyperparameter | CIFAR-10/100 | Tiny-ImageNet |
|---|---|---|
| Learning Rate | $3 \cdot 10^{-4}$ | $10^{-5}$ |
| Minimum Learning Rate | $10^{-6}$ | $10^{-6}$ |
| Label Smoothing | 0.1 | 0.1 |
| Epochs | 200 | 200 |

The augmentation pipeline follows the same setup used to train the base VGG and ViT models; see Appendix E for details. All code for reproduction is available in our open-source repository.

### F.4. Neuron Importance Score Details

#### F.4.1. IMPLEMENTATION

Neuron importance scores are computed using the `LayerConductance` (Dhamdhere et al., 2019) and `LayerDeepLIFT` (Shrikumar et al., 2017) implementations from Captum (Kokhlikyan et al., 2020). Our fusion framework is not restricted to these methods and supports any importance scoring approach.

#### F.4.2. COMPUTATION

Importance scores can be estimated either (i) independently by each model using its own training or validation data, or (ii) jointly using the fusion dataset prior to fusion. The first approach typically yields more reliable estimates and, in our experiments, produces higher zero-shot accuracy. It also aligns naturally with the federated learning setting, where clients can compute scores locally and transmit them alongside their models.

For our benchmarks, we adopt the first approach, as it generally provides more accurate scores, particularly when the fusion dataset is skewed. In the full-dataset setup, both approaches perform comparably.

#### F.4.3. SCORE SELECTION

The choice of importance score can be treated as a hyperparameter, analogous to learning rate or weight decay, and can in principle be selected via standard procedures such as cross-validation. In our experiments, uniform scores rarely outperformed Conductance or DeepLIFT. The latter two performed comparably overall, with Conductance showing a slight advantage in some settings.

### F.5. Algorithm Runtime Comparison

We report runtime measurements for concrete instantiations of our fusion framework rather than benchmarking it in the abstract. This is also the appropriate operational view: model fusion is typically a *one-off* procedure used to combine *independently trained* networks using a modest fusion dataset. Unlike federated learning or distributed training, fusion is not executed repeatedly in a training loop and does not require frequent synchronization or rebroadcasting. As a result, fusion can be moderately slower than lightweight baselines while remaining fully practical, provided it yields qualitatively better fused models.

**VGG Fusion Runtime.** We evaluate fusion runtimes for VGG networks on CIFAR-10 in Table 10. All methods fuse the same pair of models using an identical number of fusion samples per iteration; runtimes are averaged over multiple runs. As expected, our KF-based methods are slower than lightweight baselines such as OTFusion and Git Re-Basin, reflecting the additional computation required for richer matching and neuron-level interpolation.

**ViT Fusion Runtime.** For Vision Transformers, we report fusion runtimes on CIFAR-100 using KF-Gradient with uniform importance scores in Table 11. Runtime scales predictably with the number of fusion samples, remaining practical even for large fusion datasets.

*Table 10.* **Algorithm runtime** comparison when fusing VGG networks on CIFAR-10. We fused the same two models 10 times and averaged the run times. All algorithms were run with the same 400 samples in each iteration. Fusion was done on a single NVIDIA RTX A5000 GPU.

| Algorithm | Runtime (seconds) |
|---|---|
| OT Fusion | 0.7 |
| Git Re-Basin | 1.0 |
| HF-Linear Uniform (Ours) | 14.2 |
| KF-Linear Uniform (Ours) | 78.3 |
| KF-Gradient Uniform (Ours) | 16.5 |

*Table 11.* **Algorithm runtime** comparison when fusing ViT networks on CIFAR-100. We fused the same two models 5 times using our KF-Gradient method with uniform importance scores and averaged the run times.

| Fusion Samples | Runtime (s) |
|---|---|
| 400 | 38.1 |
| 6000 | 632.5 |

**Runtime considerations across problem settings.** Although our fusion procedures are more computationally demanding than prior baselines, the relevant question is whether the additional cost enables capabilities that baselines cannot provide.

*Non-IID / class-sharded data: enabling a previously unmet capability.* In highly non-IID or class-sharded regimes, existing fusion methods frequently yield fused models with near-random zero-shot accuracy, rendering them unusable without extensive fine-tuning. Our methods maintain strong zero-shot performance in precisely these settings. The additional fusion compute therefore does not represent a mere constant-factor slowdown âĂŤ it enables *practical zero-shot fusion under distribution shift*, a setting in which competing approaches largely fail. Runtime comparisons in isolation can thus be misleading: when alternative methods do not produce a viable fused model, the effective cost is not "faster fusion" but "no solution."

*Full-dataset settings: fusion cost is dominated by fine-tuning.* In full-dataset scenarios, the dominant computational expense is fine-tuning rather than fusion, as made explicit in Table 12. This table reports an end-to-end runtime breakdown for ViT fusion on Tiny-ImageNet under a 100-epoch fine-tuning budget. Our fusion step adds 602 seconds to a baseline fine-tuning cost of 8717 seconds—an overhead of less than 10%. Moreover, since our fused models already achieve high zero-shot accuracy, they typically require *fewer* fine-tuning epochs to reach a target accuracy, further reducing practical end-to-end cost.

*Table 12.* **End-to-end runtime breakdown** for ViT fusion on Tiny-ImageNet. Fine-tuning is performed for 100 epochs. All runtimes are averaged over five runs and measured in seconds.

| Algorithm | Fusion Time | Fine-Tuning Time (100 Epochs) | Total Runtime |
|---|---|---|---|
| Theoretical 0-Time Fusion | 0 | 8717 ($\pm$40) | 8717 |
| Ours (No Fine-Tuning) | 602 ($\pm$34) | 0 | 603 |
| Ours (With Fine-Tuning) | 602 ($\pm$34) | 8717 ($\pm$40) | 9319 |

**Potential speedups.** If fusion latency is critical, we note that: (i) target computations are independent across network levels and can be parallelized, (ii) fusion scales naturally with additional hardware, and (iii) the present codebase prioritizes clarity and flexibility over speed, and has not been specifically engineered for performance.

Overall, while our methods are slower than lightweight fusion baselines, this additional compute is (i) essential for strong zero-shot fusion in non-IID and class-sharded regimes where prior methods are ineffective, and (ii) negligible relative to fine-tuning cost in standard full-dataset pipelines.

# G. Additional Results

In this section, besides complementary tables, we will also present the full tables for results shown earlier. These full tables include the standard deviation for base models, as well as the performance of each fusion algorithm for each neuron importance score.

## G.1. VGGs on CIFAR-10

*Table 13.* **Test accuracy** comparison when fusing VGG11 networks on CIFAR-10 for **Non-IID** splits. Fusion was performed using 400 data points sampled from the dataset seen by the first model. The same fusion data was used for all algorithms. Results are averaged over 5 seeds. For KD, the student model was randomly initialized, whereas for LP, it was initialized with the weights of the first base model.

| Method | 2-WAY SPLIT | 4-WAY SPLIT | 8-WAY SPLIT |
|---|---|---|---|
| Individual Models | $83.8_{\pm2.7}$, $77.3_{\pm2.1}$ | $79.8_{\pm3.2}$, $77.8_{\pm2.9}$, $74.7_{\pm3.3}$, $69.7_{\pm4.6}$ | $72.3_{\pm1.8}$, $70.6_{\pm2.4}$, $67.7_{\pm1.1}$, $66.1_{\pm1.9}$ $65.4_{\pm1.6}$, $63.4_{\pm1.2}$, $58.6_{\pm3.5}$, $55.5_{\pm3.0}$ |
| Ensemble | $89.1_{\pm0.4}$ | $85.7_{\pm0.2}$ | $78.9_{\pm0.8}$ |
| Vanilla Averaging | $11.5_{\pm1.5}$ | $10.0_{\pm0.0}$ | $10.0_{\pm0.0}$ |
| KD | $38.4_{\pm3.3}$ | $37.8_{\pm2.6}$ | $42.4_{\pm2.9}$ |
| LP | $85.5_{\pm2.0}$ | $\mathbf{81.4}_{\pm1.6}$ | $\mathbf{73.8}_{\pm1.2}$ |
| OTF Uniform | $49.6_{\pm6.5}$ | $14.8_{\pm2.4}$ | $11.3_{\pm2.3}$ |
| OTF Conductance | $40.6_{\pm3.9}$ | $11.7_{\pm2.4}$ | $11.0_{\pm1.0}$ |
| OTF DeepLIFT | $39.4_{\pm7.3}$ | $13.4_{\pm3.9}$ | $13.4_{\pm1.0}$ |
| Git Re-Basin[1] Uniform | $55.7_{\pm4.6}$ | N/A | N/A |
| Git Re-Basin Conductance | $73.8_{\pm2.3}$ | N/A | N/A |
| Git Re-Basin DeepLIFT | $76.6_{\pm3.8}$ | N/A | N/A |
| HF-Linear Uniform (Ours) | $76.7_{\pm2.1}$ | N/A | N/A |
| HF-Linear Conductance (Ours) | $86.4_{\pm0.6}$ | N/A | N/A |
| HF-Linear DeepLIFT (Ours) | $86.3_{\pm0.6}$ | N/A | N/A |
| KF-Linear Uniform (Ours) | $85.5_{\pm1.3}$ | $78.8_{\pm1.0}$ | $69.5_{\pm1.7}$ |
| KF-Linear Conductance (Ours) | $\mathbf{86.6}_{\pm0.7}$ | $79.5_{\pm0.8}$ | $71.4_{\pm1.4}$ |
| KF-Linear DeepLIFT (Ours) | $86.5_{\pm0.7}$ | $79.7_{\pm0.7}$ | $71.6_{\pm1.4}$ |
| HF-Gradient Uniform (Ours) | $76.3_{\pm1.6}$ | N/A | N/A |
| HF-Gradient Conductance (Ours) | $84.8_{\pm1.3}$ | N/A | N/A |
| HF-Gradient DeepLIFT (Ours) | $84.9_{\pm1.2}$ | N/A | N/A |
| KF-Gradient Uniform (Ours) | $84.3_{\pm1.3}$ | $76.6_{\pm1.4}$ | $67.0_{\pm1.5}$ |
| KF-Gradient Conductance (Ours) | $84.9_{\pm1.3}$ | $77.8_{\pm1.0}$ | $69.4_{\pm1.4}$ |
| KF-Gradient DeepLIFT (Ours) | $84.7_{\pm1.2}$ | $77.8_{\pm0.8}$ | $69.0_{\pm1.2}$ |

---

[1]Git Re-Basin reduces to OTFusion when solving the OT problem exactly with uniform importance scores. In practice, OTFusion uses preactivations (Singh & Jaggi, 2020), while Git Re-Basin uses activations (Ainsworth et al., 2023); we follow these defaults. Empirically, both yield the same fused model when using preactivations.

*Table 14*. **Test accuracy** comparison when fusing VGG11 networks on CIFAR-10 for **Sharded** splits. Fusion was performed using 400 data points sampled from the dataset seen by the first model. The same fusion data was used for all algorithms. Results are averaged over 5 seeds. For KD, the student model was randomly initialized, whereas for LP, it was initialized with the weights of the first base model.

| Method | 2-WAY SPLIT | 4-WAY SPLIT | 6-WAY SPLIT |
|---|---|---|---|
| Individual Models | $47.8_{\pm 0.5}$, $46.7_{\pm 0.8}$ | $29.1_{\pm 0.1}$, $28.8_{\pm 0.2}$, $19.7_{\pm 0.2}$, $19.1_{\pm 0.6}$ | $19.9_{\pm 0.0}$, $19.8_{\pm 0.1}$, $19.5_{\pm 0.2}$, $15.2_{\pm 3.7}$, $10.0_{\pm 0.0}$, $10.0_{\pm 0.0}$ |
| Ensemble | $80.2_{\pm 2.3}$ | $58.3_{\pm 1.7}$ | $41.3_{\pm 1.7}$ |
| Vanilla Averaging | $12.1_{\pm 2.2}$ | $10.0_{\pm 0.0}$ | $10.0_{\pm 0.0}$ |
| KD | $30.2_{\pm 3.3}$ | $22.8_{\pm 2.6}$ | $17.4_{\pm 4.3}$ |
| LP | $43.2_{\pm 3.0}$ | $15.4_{\pm 2.4}$ | $13.3_{\pm 1.7}$ |
| ZipIt! | $20.9_{\pm 6.5}$ | N/A | N/A |
| OTF Uniform | $24.4_{\pm 6.0}$ | $10.6_{\pm 0.8}$ | $10.0_{\pm 0.0}$ |
| OTF Conductance | $22.5_{\pm 7.9}$ | $11.4_{\pm 2.1}$ | $10.0_{\pm 0.0}$ |
| OTF DeepLIFT | $28.1_{\pm 7.8}$ | $11.8_{\pm 4.8}$ | $10.0_{\pm 0.1}$ |
| Git Re-Basin Uniform | $27.8_{\pm 6.0}$ | N/A | N/A |
| Git Re-Basin Conductance | $60.5_{\pm 3.5}$ | N/A | N/A |
| Git Re-Basin DeepLIFT | $63.8_{\pm 6.9}$ | N/A | N/A |
| HF-Linear Uniform (Ours) | $51.9_{\pm 1.7}$ | N/A | N/A |
| HF-Linear Conductance (Ours) | $78.9_{\pm 1.9}$ | N/A | N/A |
| HF-Linear DeepLIFT (Ours) | $79.0_{\pm 2.0}$ | N/A | N/A |
| KF-Linear Uniform (Ours) | $78.6_{\pm 0.7}$ | $52.9_{\pm 2.7}$ | $35.8_{\pm 2.4}$ |
| KF-Linear Conductance (Ours) | $\mathbf{80.9}_{\pm 1.9}$ | $\mathbf{56.4}_{\pm 2.0}$ | $\mathbf{40.3}_{\pm 1.2}$ |
| KF-Linear DeepLIFT (Ours) | $\mathbf{80.9}_{\pm 1.7}$ | $56.2_{\pm 1.8}$ | $40.1_{\pm 1.3}$ |
| HF-Gradient Uniform (Ours) | $50.7_{\pm 3.6}$ | N/A | N/A |
| HF-Gradient Conductance (Ours) | $74.3_{\pm 1.3}$ | N/A | N/A |
| HF-Gradient DeepLIFT (Ours) | $74.4_{\pm 1.3}$ | N/A | N/A |
| KF-Gradient Uniform (Ours) | $72.6_{\pm 2.3}$ | $45.8_{\pm 2.9}$ | $33.8_{\pm 1.9}$ |
| KF-Gradient Conductance (Ours) | $74.0_{\pm 1.8}$ | $47.9_{\pm 3.4}$ | $34.8_{\pm 2.5}$ |
| KF-Gradient DeepLIFT (Ours) | $73.2_{\pm 1.9}$ | $46.3_{\pm 4.5}$ | $34.7_{\pm 1.5}$ |

*Table 15.* **Test accuracy** comparison when fusing VGG11 networks pairwise on CIFAR-10 trained on the **full dataset**. Results are averaged across 3 seeds. Fusion was performed using 400 samples from the full dataset. The same fusion data was used for all algorithms. Fine-tuning was performed for 200 epochs with a learning rate of $3 \cdot 10^{-4}$ and a cosine annealing with warm restarts scheduler with a minimum learning rate of $10^{-6}$.

| **Method** | ZERO-SHOT | FINETUNED |
|---|---|---|
| Individual Models | $93.2_{\pm 0.1}$ $93.0_{\pm 0.1}$ | $93.2_{\pm 0.1}$ $93.2_{\pm 0.1}$ |
| Vanilla Averaging | $9.3_{\pm 1.4}$ | $-$ |
| Ensemble | $94.1_{\pm 0.1}$ | $94.2_{\pm 0.1}$ |
| OTF Uniform | $72.6_{\pm 5.2}$ | $93.2_{\pm 0.2}$ |
| OTF Conductance | $46.7_{\pm 11.1}$ | $93.5_{\pm 0.3}$ |
| OTF DeepLIFT | $49.8_{\pm 4.0}$ | $93.4_{\pm 0.1}$ |
| Git Re-Basin Uniform | $77.1_{\pm 3.6}$ | $93.5_{\pm 0.1}$ |
| Git Re-Basin Conductance | $61.8_{\pm 2.8}$ | $93.3_{\pm 0.1}$ |
| Git Re-Basin DeepLIFT | $65.3_{\pm 5.9}$ | $\mathbf{93.6}_{\pm 0.0}$ |
| HF-Linear Uniform (Ours) | $87.0_{\pm 0.2}$ | $93.3_{\pm 0.1}$ |
| HF-Linear Conductance (Ours) | $74.3_{\pm 1.8}$ | $93.3_{\pm 0.1}$ |
| HF-Linear DeepLIFT (Ours) | $73.4_{\pm 1.7}$ | $93.4_{\pm 0.2}$ |
| KF-Linear Uniform (Ours) | $74.2_{\pm 0.8}$ | $93.2_{\pm 0.2}$ |
| KF-Linear Conductance (Ours) | $74.9_{\pm 1.4}$ | $93.4_{\pm 0.1}$ |
| KF-Linear DeepLIFT (Ours) | $75.3_{\pm 0.5}$ | $93.4_{\pm 0.2}$ |
| HF-Gradient Uniform (Ours) | $\mathbf{88.1}_{\pm 0.2}$ | $93.4_{\pm 0.1}$ |
| HF-Gradient Conductance (Ours) | $87.9_{\pm 0.2}$ | $93.4_{\pm 0.1}$ |
| HF-Gradient DeepLIFT (Ours) | $88.1_{\pm 0.1}$ | $93.3_{\pm 0.3}$ |
| KF-Gradient Uniform (Ours) | $85.0_{\pm 0.2}$ | $93.0_{\pm 0.2}$ |
| KF-Gradient Conductance (Ours) | $85.9_{\pm 0.7}$ | $93.2_{\pm 0.1}$ |
| KF-Gradient DeepLIFT (Ours) | $86.2_{\pm 1.3}$ | $93.0_{\pm 0.1}$ |

*Table 16.* Fusing VGG11s trained on two non-IID splits of CIFAR-10. We vary the number of parameters in each layer of the base models by 0.5x, 1x, 2x or 4x. The fused model has the same width as Model 0.

| | | | | KF-Linear | | | KF-Gradient | | | | |
|---|---|---|---|---|---|---|---|---|---|---|---|
| Model 0 Width | Model 1 Width | Model 0 | Model 1 | Uniform | Conductance | DeepLIFT | Uniform | Conductance | DeepLIFT | OT Fusion | KD |
| 1x | 1x | 87.1 | 74.4 | 85.8 | 86.4 | 86.7 | 87.8 | **88.0** | 87.8 | 50.6 | 84.9 |
| 0.5x | 0.5x | 85.6 | 72.2 | 81.8 | 83.5 | 83.0 | 85.9 | 85.9 | **85.9** | 52.0 | 82.1 |
| 2x | 2x | 88.3 | 74.6 | 87.4 | 88.0 | 88.0 | 89.1 | 89.0 | **89.1** | 50.2 | 85.4 |
| 4x | 4x | 88.5 | 74.9 | 87.3 | 88.1 | 88.2 | **89.1** | 89.1 | 89.1 | 49.7 | 85.7 |
| 1x | 0.5x | 87.1 | 72.2 | 85.7 | 87.0 | 86.6 | 88.0 | **88.1** | 87.9 | 56.6 | 84.7 |
| 1x | 2x | 87.1 | 74.6 | 85.1 | 86.1 | 86.4 | 87.7 | 87.5 | **87.8** | 29.8 | 84.6 |
| 1x | 4x | 87.1 | 74.9 | 84.9 | 86.2 | 86.3 | 87.8 | 87.7 | **87.8** | 18.6 | 85.0 |

## G.2. ViTs on CIFAR-100 and Tiny-ImageNet

### G.2.1. CIFAR-100

*Table 17.* **Test accuracy** comparison when fusing ViT networks on CIFAR-100 for **Sharded** splits. Fusion was performed using 5000 data points sampled from the dataset seen by the first model. For activations-based (acts) Transformer OTFusion, we used a subset of 200 samples[*]. The weights-based variant (wts) does not use data. Results are averaged over 5 seeds. For KD, the student model was randomly initialized, whereas for LP, it was initialized with the weights of the first base model. This table is complementary to Table 2.

| Method | 2-WAY SPLIT | 4-WAY SPLIT | | 6-WAY SPLIT | | |
|---|---|---|---|---|---|---|
| Individual Models | $38.6_{\pm0.5}$ $37.2_{\pm0.6}$ | $20.4_{\pm0.2}$ $19.5_{\pm0.2}$ | $19.9_{\pm0.1}$ $19.2_{\pm0.3}$ | $14.3_{\pm0.2},$ $13.2_{\pm0.2}$ | $13.7_{\pm0.3}$ $12.8_{\pm0.3}$ | $13.5_{\pm0.2}$ $12.2_{\pm0.6}$ |
| Ensemble | $63.7_{\pm0.4}$ | $53.4_{\pm1.8}$ | | $45.4_{\pm2.0}$ | | |
| Vanilla Averaging | $2.2_{\pm0.6}$ | $1.4_{\pm0.2}$ | | $1.1_{\pm0.3}$ | | |
| KD | $41.0_{\pm1.5}$ | $\mathbf{47.0}_{\pm0.8}$ | | $\mathbf{43.6}_{\pm0.8}$ | | |
| LP | $45.3_{\pm1.2}$ | $33.9_{\pm1.0}$ | | $27.2_{\pm1.0}$ | | |
| Transformer OTFusion acts Uniform | $2.2_{\pm0.4}$ | $1.2_{\pm0.2}$ | | $1.0_{\pm0.0}$ | | |
| Transformer OTFusion acts Conductance | $2.3_{\pm0.3}$ | $1.1_{\pm0.3}$ | | $1.0_{\pm0.0}$ | | |
| Transformer OTFusion acts DeepLIFT | $2.3_{\pm0.5}$ | $1.0_{\pm0.0}$ | | $1.0_{\pm0.0}$ | | |
| Transformer OTFusion wts Uniform | $3.9_{\pm0.8}$ | $1.5_{\pm0.4}$ | | $1.2_{\pm0.3}$ | | |
| Transformer OTFusion wts Conductance | $4.4_{\pm1.2}$ | $1.3_{\pm0.3}$ | | $1.3_{\pm0.4}$ | | |
| Transformer OTFusion wts DeepLIFT | $4.1_{\pm1.2}$ | $1.2_{\pm0.3}$ | | $1.0_{\pm0.0}$ | | |
| HF-Gradient Uniform (Ours) | $48.4_{\pm0.9}$ | N/A | | N/A | | |
| HF-Gradient Conductance (Ours) | $54.5_{\pm1.2}$ | N/A | | N/A | | |
| HF-Gradient DeepLIFT (Ours) | $\mathbf{54.5}_{\pm1.2}$ | N/A | | N/A | | |
| KF-Gradient Uniform (Ours) | $54.2_{\pm1.3}$ | $41.3_{\pm1.0}$ | | $34.2_{\pm1.5}$ | | |
| KF-Gradient Conductance (Ours) | $54.4_{\pm1.1}$ | $40.9_{\pm1.2}$ | | $34.7_{\pm1.3}$ | | |
| KF-Gradient DeepLIFT (Ours) | $54.4_{\pm1.1}$ | $41.4_{\pm1.0}$ | | $34.4_{\pm1.6}$ | | |

[*] For Transformer OTFusion, we use 200 fusion samples following Imfeld et al. (2024), who report saturating accuracy at this scale for ViT architectures (Figure 5 therein). Attempts to scale to 5000 samples resulted in out-of-memory errors with the authors' implementation

*Table 18.* **Test accuracy** comparison when fusing ViT networks on CIFAR-100 trained on the **full dataset**. Results for 2-way fusion are averaged over 3 seeds, while results for 4-way fusion are shown only for a single seed. Fusion was performed with 5000 samples from the full dataset, except for activations-based Transformer OTFusion, which used 200 samples[*]. Fine-tuning was performed for 200 epochs with a learning rate of $3 \cdot 10^{-4}$ and a cosine annealing with warm restarts scheduler with a minimum learning rate of $10^{-6}$. During fine-tuning, the base models failed to improve. This table is complementary to Table 5.

| Method | 2-WAY ZERO-SHOT | 2-WAY FINETUNED | 4-WAY ZERO-SHOT | 4-WAY FINETUNED |
|---|---|---|---|---|
| Individual Models | $\mathbf{73.9}_{\pm 0.2}$ $73.4_{\pm 0.3}$ | $73.5_{\pm 0.3}$ $73.0_{\pm 0.3}$ | $\mathbf{74.1}$, 73.6, 73.0, 72.9 | 73.7, 73.2, 72.7, 72.7 |
| Ensemble Vanilla Averaging | $75.7_{\pm 0.3}$ $1.9_{\pm 0.2}$ | $75.5_{\pm 0.4}$ − | 76.6 1.1 | 76.4 − |
| Transf. OTF acts Uniform | $2.7_{\pm 0.2}$ | $73.8_{\pm 0.4}$ | 1.0 | 63.7 |
| Transf. OTF acts Conductance | $2.4_{\pm 0.8}$ | $73.7_{\pm 0.4}$ | 1.0 | 63.0 |
| Transf. OTF acts DeepLIFT | $2.3_{\pm 0.9}$ | $74.0_{\pm 0.4}$ | 1.0 | 62.6 |
| Transf. OTF wts Uniform | $4.3_{\pm 0.2}$ | $74.0_{\pm 0.4}$ | 1.0 | 72.6 |
| Transf. OTF wts Conductance | $3.2_{\pm 1.1}$ | $73.8_{\pm 0.3}$ | 1.0 | 68.6 |
| Transf. OTF wts DeepLIFT | $3.2_{\pm 1.5}$ | $73.9_{\pm 0.3}$ | 1.0 | 68.8 |
| HF-Gradient Uniform (Ours) | $57.0_{\pm 1.1}$ | $74.8_{\pm 0.4}$ | N/A | N/A |
| HF-Gradient Conductance (Ours) | $58.6_{\pm 1.1}$ | $75.0_{\pm 0.5}$ | N/A | N/A |
| HF-Gradient DeepLIFT (Ours) | $58.6_{\pm 1.3}$ | $75.2_{\pm 0.6}$ | N/A | N/A |
| KF-Gradient Uniform (Ours) | $63.0_{\pm 1.2}$ | $75.2_{\pm 0.5}$ | 57.5 | 75.2 |
| KF-Gradient Conductance (Ours) | $62.8_{\pm 0.9}$ | $\mathbf{75.4}_{\pm 0.1}$ | 57.5 | 75.2 |
| KF-Gradient DeepLIFT (Ours) | $62.4_{\pm 1.9}$ | $75.2_{\pm 0.1}$ | 57.1 | $\mathbf{75.6}$ |

### G.2.2. TINY-IMAGENET

*Table 19.* **Test accuracy** comparison when fusing ViT networks on Tiny-ImageNet for **Sharded 2-way splits**. Fusion was performed using 5000 data points sampled from the dataset seen by the first model. For activations-based (acts) Transformer OTFusion, we used a subset of 200 samples[*]. The weights-based variant (wts) does not use data. Results are averaged over 5 seeds. For KD, the student model was randomly initialized, whereas for LP, it was initialized with the weights of the first base model.

| Method | 2-WAY SPLIT |
|---|---|
| Individual Model 0 | $28.3_{\pm 0.2}$ |
| Individual Model 1 | $27.5_{\pm 0.5}$ |
| Ensemble | $44.5_{\pm 0.4}$ |
| Vanilla Averaging | $0.9_{\pm 0.4}$ |
| KD | $11.3_{\pm 1.4}$ |
| LP | $30.8_{\pm 0.6}$ |
| Transformer OTFusion acts Uniform | $2.6_{\pm 1.2}$ |
| Transformer OTFusion acts Conductance | $2.7_{\pm 1.6}$ |
| Transformer OTFusion acts DeepLIFT | $2.6_{\pm 1.6}$ |
| Transformer OTFusion wts Uniform | $4.0_{\pm 0.9}$ |
| Transformer OTFusion wts Conductance | $2.3_{\pm 0.9}$ |
| Transformer OTFusion wts DeepLIFT | $2.1_{\pm 1.1}$ |
| HF-Gradient Uniform (Ours) | $30.6_{\pm 0.8}$ |
| HF-Gradient Conductance (Ours) | $32.4_{\pm 0.9}$ |
| HF-Gradient DeepLIFT (Ours) | $\mathbf{32.5}_{\pm 1.1}$ |
| KF-Gradient Uniform (Ours) | $32.3_{\pm 0.9}$ |
| KF-Gradient Conductance (Ours) | $32.4_{\pm 0.6}$ |
| KF-Gradient DeepLIFT (Ours) | $31.5_{\pm 1.1}$ |

*Table 20.* **Test accuracy** comparison when fusing ViT networks on Tiny-ImageNet trained on the **full dataset**. Results for 2-way fusion are averaged over 2 seeds. Fusion was performed with 5000 samples from the full dataset, except for activations-based Transformer OTFusion, which used 200 samples[*]. Fine-tuning was performed for 200 epochs with a learning rate of $10^{-5}$ and a cosine annealing with warm restarts scheduler with a minimum learning rate of $10^{-6}$. This table is complementary to Table 6.

| Method | 2-WAY ZERO-SHOT | 2-WAY FINETUNED |
|---|---|---|
| Individual Models | $\mathbf{52.7}_{\pm 0.2}$ | $53.2_{\pm 0.5}$ |
|  | $51.7_{\pm 0.0}$ | $51.7_{\pm 0.0}$ |
| Ensemble | $54.9_{\pm 0.4}$ | $55.7_{\pm 0.4}$ |
| Vanilla Averaging | $1.1_{\pm 0.1}$ | − |
| Transf. OTF acts Uniform | $1.4_{\pm 0.1}$ | $53.6_{\pm 0.1}$ |
| Transf. OTF acts Conductance | $1.4_{\pm 0.3}$ | $53.7_{\pm 0.2}$ |
| Transf. OTF acts DeepLIFT | $1.2_{\pm 0.3}$ | $53.7_{\pm 0.2}$ |
| Transf. OTF wts Uniform | $3.1_{\pm 0.2}$ | $53.8_{\pm 0.1}$ |
| Transf. OTF wts Conductance | $2.2_{\pm 0.8}$ | $53.6_{\pm 0.1}$ |
| Transf. OTF wts DeepLIFT | $2.1_{\pm 0.9}$ | $53.7_{\pm 0.3}$ |
| HF-Gradient Uniform (Ours) | $40.5_{\pm 1.5}$ | $53.0_{\pm 0.3}$ |
| HF-Gradient Conductance (Ours) | $42.1_{\pm 4.9}$ | $53.6_{\pm 0.2}$ |
| HF-Gradient DeepLIFT (Ours) | $41.9_{\pm 3.2}$ | $53.7_{\pm 0.1}$ |
| KF-Gradient Uniform (Ours) | $42.5_{\pm 0.5}$ | $53.9_{\pm 0.1}$ |
| KF-Gradient Conductance (Ours) | $42.6_{\pm 0.8}$ | $\mathbf{54.2}_{\pm 0.4}$ |
| KF-Gradient DeepLIFT (Ours) | $42.9_{\pm 0.3}$ | $53.8_{\pm 0.8}$ |

## H. Existing Assets and Licenses

We make use of code from the following sources:

1. OTFusion (Singh & Jaggi, 2020), Open-Source, https://github.com/sidak/otfusion.

2. ViT-CIFAR (omihub777), MIT License, https://github.com/omihub777/ViT-CIFAR/blob/main/LICENSE.

3. Captum (Kokhlikyan et al., 2020), BSD 3-Clause License, https://github.com/pytorch/captum/blob/master/LICENSE.

4. ffcv (Leclerc et al., 2022), Apache License 2.0, https://github.com/libffcv/ffcv-imagenet/blob/main/LICENSE

