# OpenReview forum: "Model Fusion via Retrofitting"
_ICML.cc/2026/Conference — ICML 2026 regular_

### Official Review · Reviewer_DYD2 · 2026-02-26

**Soundness:** 3
**Presentation:** 3
**Significance:** 3
**Originality:** 3
**Overall Recommendation:** 4
**Confidence:** 3

**Summary:**

This paper proposes a neuron-centric model fusion framework that introduces neuron importance scores and formulates fusion as an importance-weighted representation matching problem. The approach naturally decouples into two stages: neuron clustering and layer-wise reconstruction. Depending on how many networks are fused, the clustering stage uses either K-means or the Hungarian algorithm and supports many-to-one matching. For reconstruction, the method provides closed-form solutions for linear layers and uses SGD to optimize nonlinear layers, making it broadly applicable and largely architecture-agnostic. Experiments across non-IID, sharded, and full-data settings show stable performance. In zero-shot fusion, the method outperforms several baselines and in some cases approaches ensemble performance.

**Compliance With Llm Reviewing Policy:**

Affirmed.

**Final Justification:**

Thank you for the rebuttal. The authors have clarified several points, including the definition of the zero-shot setting and the role of the fusion dataset, and provided additional experimental evidence.

However, some concerns remain, particularly regarding the dependency on fusion data and the limited analysis of optimization stability for nonlinear layers. While the rebuttal improves clarity, these issues are not fully resolved.

Overall, my assessment remains unchanged. I maintain my score.

**Key Questions For Authors:**

1. How exactly is “zero-shot” defined in this work? Is fusion data used in this setting? If so, it would be helpful to include ablations on the two-stage decoupling to quantify their individual contributions.

2. How sensitive is the method to the size and distribution of the fusion dataset? While experiments with different sample sizes are provided, it would be useful to clarify the minimum viable amount of fusion data. Additionally, what happens when the fusion data distribution significantly shifts from the base training distribution?

3. In strongly nonlinear and deep networks, how is optimization stability ensured? Has sensitivity to initialization been systematically evaluated? Is there evidence of error accumulation in deeper models?

4. The paper states that the code has been open-sourced, but I could not find the public repository or access link. Could the authors clarify where the code is available?

**Limitations:**

yes

**Strengths And Weaknesses:**

Strengths:

1. The motivation is clear and reasonably novel. The paper systematically incorporates neuron importance into the model fusion framework and unifies it under a neuron-centric representation coverage objective. Compared with traditional methods such as OTFusion, this design better reflects the realistic assumption that neurons contribute unequally in neural networks.

2. The two-stage formulation is expressive and flexible. It supports many-to-one matching and models with different widths, and can be applied to arbitrary differentiable layers, indicating good generality.

3. The method performs strongly under extreme non-IID and sharded settings. In zero-shot scenarios it clearly outperforms traditional OTFusion-style approaches.

Weaknesses:

1. The method appears to rely heavily on fusion data, and performance seems sensitive to the scale and distribution of the fusion dataset.

2. The zero-shot setting is somewhat ambiguous, and the necessity of the two-stage decoupling is not directly validated through ablation.

3. Optimization of nonlinear layers may face stability and initialization sensitivity issues. Although the paper includes some engineering mitigations, deeper analysis of stability or failure cases is missing.

---

> ### Author Rebuttal · Authors · 2026-03-31
>
> We thank you for the thoughtful and constructive feedback, and for recognizing the novelty of our formulation, its flexibility, and strong performance in non-IID and zero-shot settings.
>
> Below, we first respond to the weaknesses, followed by answers to the specific questions.
>
> &nbsp;
>
> **Weaknesses**
>
> 1. We agree that our fusion method is reliant on the quality of the fusion data. Nonetheless, we argue this is a fundamental characteristic of activation-based methods, including ZipIt! [1] and MuDSC [2] that other reviewers have mentioned. Furthermore, existing methods that work directly with weights such as weights-based OTFusion [3, 4] require fine tuning which is reliant on the availability of a fusion (or fine tuning) dataset
>
> 2. We clarify zero-shot fusion in our response to Question 1.
>
> 3. We agree that optimizing nonlinear layers is difficult in general (see also our response to Question 3). Nonetheless, we believe that this can be mitigated to some extent with larger training datasets as we show empirically. Regarding initialization, we have experimented with random initializations, and we did not find any difference in practice.
>
> &nbsp;
>
> **Questions**
>
> 1. In our work, zero-shot means that further fine-tuning is not done after fusion. This is as opposed to one-shot fusion which we define to be fusion done with a further fine-tuning step going through the full training data of all base models again. Our method uses a small proxy 'fusion dataset' to align activations, but once the layers are fused, no additional training is performed. We agree that an ablation for whether we perform the two stage decomposition is sensible, and we hypothesize that this may even lead to improvements for our K-means based versions in some settings. In the current work, we had chosed to focus on the two step approach as it is necessitated by the Hungarian matching variants of our methods to simplify the simultaneous discussion of both methods.
>
> 2. We provide an additional experiment with ImageNet, as done in Zipit! [1], where base models were trained on splits of 200 classes and then fused. Here, we vary the fusion dataset size for KF-gradient (uniform importance scores):
>
>
> | Algorithm                               | Joint         | Task A        | Task B           |
> |-----------------------------------------|------------------|------------------|------------------|
> | Model A                                 | 41.1 (± 0.1)     | 82.3 (± 0.2)     | 0.0 (± 0.0)      |
> | Model B                                 | 41.1 (± 0.6)     | 0.0 (± 0.0)      | 82.1 (± 1.1)     |
> | ZipIt! (25k samples)                  | 37.7 (± 1.7)     | 37.1 (± 4.0)     | 38.3 (± 1.4)     |
> | KF-gradient (1k samples)        | 25.1 (± 0.7)     | 9.8 (± 0.8)      | 40.4 (± 0.8)     |
> | KF-gradient (2k samples)        | 34.7 (± 2.6)     | 17.4 (± 3.1)     | 51.9 (± 2.1)     |
> | KF-gradient (5k samples)        | 49.2 (± 1.0)     | 33.4 (± 1.4)     | 64.9 (± 1.3)     |
> | KF-gradient (10k samples)       | 54.8 (± 0.8)     | 40.3 (± 1.6)     | 69.3 (± 1.0)     |
> | **KF-gradient (25k samples)**   | **58.9 (± 1.0)** | **46.1 (± 1.4)** | **71.7 (± 0.9)** |
> | Ensemble                                | 62.4 (± 1.0)     | 47.7 (± 1.7)     | 77.1 (± 1.2)     |
>
>
> In general, it is difficult to provide an estimate of the minimum data required a priori as we empirically observe that it depends on both the model architecture and the task being performed. Compare, for example, the above experiment with the BloodMNIST experiment in Table 3 using only 400 samples from a subset of the classes.
>
> 3. We agree that our methods are unable to guarantee optimization stability in the general case. Indeed, this is one of the motivations for performing the fusion in a layerwise manner and decoupling the error for each layer - It allows us to fix the targets individually by layer to eliminate neuron cluster assignments as an additional source of training instability.
>
> 4. We plan to open-source our implementation if our paper gets accepted. However, we have asked the AC if it's possible to share an anonymized repo, so that you can take a look during the reviewing period if you wish.
>
> &nbsp;
>
> ---
>
> **References**
>
> [1] Stoica, G., Bolya, D., Bjorner, J., Ramesh, P., Hearn, T. and Hoffman, J., 2023. Zipit! merging models from different tasks without training. arXiv preprint arXiv:2305.03053.
>
> [2] Xu, Z., Yuan, K., Wang, H., Wang, Y., Song, M. and Song, J., 2024. Training-free pretrained model merging. In Proceedings of the IEEE/CVF Conference on Computer Vision and Pattern Recognition (pp. 5915-5925).
>
> [3] Singh, S.P. and Jaggi, M., 2020. Model fusion via optimal transport. Advances in Neural Information Processing Systems, 33, pp.22045-22055.
>
> [4] Imfeld, M., Graldi, J., Giordano, M., Hofmann, T., Anagnostidis, S. and Singh, S.P., 2023. Transformer fusion with optimal transport. arXiv preprint arXiv:2310.05719.

---

> > ### Author Rebuttal · Reviewer_DYD2 · 2026-04-03
> >
> > Thank you for the rebuttal. The authors clarified several points, including the definition of zero-shot and provided additional experiments on fusion data size.

---

> > > ### Author Response · Authors · 2026-04-03
> > >
> > > Thank you for your remarks. Following the AC's guidance, we:
> > >
> > > &nbsp;
> > >
> > > **(a) Share anonymized code:**
> > >
> > > https://anonymous.4open.science/r/rebuttal-0157
> > >
> > > &nbsp;
> > >
> > >
> > > **(b) Clarify an implementation issue:**
> > >
> > > After submission, we identified a bug affecting the construction of the fusion dataset in the sharded data setting, leading to a discrepancy between our single-fusion and mass-fusion scripts.
> > >
> > > After fixing this issue and rerunning the experiments, the performance of our method changes only slightly (on the order of 1-2%), indicating that it is largely unaffected by this discrepancy. In contrast, some baselines (e.g., KD) are more sensitive to the construction of the fusion dataset and experience a more noticeable drop in performance.
> > >
> > > **Importantly, our method performs at least as well as before, while baseline performance decreases, leading to a larger gap.** The discussion in some experiments had to be slightly revised, but the overall conclusions remain unchanged: *our method continues to outperform the considered baselines by a larger margin.*
> > >
> > > We also evaluated an additional variant where importance scores are computed using each model's private dataset. This further improves performance, in some cases **surpassing ensemble baselines** (e.g., sharded BloodMNIST and VGG).
> > >
> > > As advised by the AC, we are unable to upload a revised manuscript, but we are happy to provide corrected tables or additional details upon request.

---

### Official Review · Reviewer_gsGv · 2026-03-09

**Soundness:** 2
**Presentation:** 2
**Significance:** 2
**Originality:** 2
**Overall Recommendation:** 3
**Confidence:** 4

**Summary:**

This paper proposes an approach to model fusion that frames the problem as representation matching. The authors introduce both linear and gradient-based variants of their algorithm, incorporating neuron saliency measures (DeepLIFT, Conductance) to guide the alignment process. Experiments are conducted on VGG11 and ViT architectures across different data partitioning scenarios.

**Compliance With Llm Reviewing Policy:**

Affirmed.

**Final Justification:**

I thank the authors for their response. However, I still believe that the evaluation is insufficient in terms of both baselines and the diversity of datasets and backbone architectures. I will remain with my original rating.

**Key Questions For Authors:**

Can you provide results on ImageNet or other large-scale datasets?

**Limitations:**

See weakness.

**Strengths And Weaknesses:**

**Strengths**

* (a) This paper is clearly written.

**Weaknesses**

* (a) Novelty of this paper is limited. The two-stage representation matching framework is essentially a refinement of existing neuron alignment methods (OTFusion, Git Re-Basin).

* (b) Unclear algorithmic details: While the authors claim a two-stage algorithm, the mathematical formulation and implementation details are poorly explained. The derivations in the appendix (e.g., page with lines 550-604) are difficult to follow with undefined notation.

* (c) Computational cost analysis missing: Model fusion efficiency is crucial for practical deployment, yet there's no discussion of time/memory complexity or wall-clock runtime comparisons. The overhead of computing saliency scores should be quantified.

* (d) Questionable fine-tuning results: Table 15 shows that base models achieve 93.2±0.1 accuracy, but after fine-tuning for 200 epochs, many methods show no improvement or even degradation. Is the learning rate of 3e-4 too high for fine-tuning? This suggests potential experimental issues.

* (e) Limited ablation studies: While different saliency methods are compared, other design choices lack thorough investigation: （1）Impact of fusion data size (only 400 samples used in Tables 13-15; (2) Relative importance of the two stages; (3) Sensitivity to different layer widths (Table 16 shows varied results with different width multipliers.

* (f) Limited experimental scope: The evaluation is restricted to only two architectures (VGG11 and ViT) on small-scale datasets. Major limitations include: (1) No evaluation on modern architectures (ResNets, EfficientNets, recent vision models); (2) No large-scale dataset evaluation (e.g., ImageNet) to demonstrate scalability; (3) Only image classification tasks - no exploration of other domains (NLP, speech) or tasks (detection, segmentation); (4) Limited to CIFAR-10/100 and Tiny-ImageNet.

---

> ### Author Rebuttal · Authors · 2026-03-31
>
> We thank you for your constructive feedback. We agree that providing experimental results on non-toy datasets will benefit the paper and we have performed an experiment we discuss below. We also take the opportunity to respond to the weaknesses raised.
>
> &nbsp;
>
> **Weaknesses**:
>
> (a) While we agree that the objective used by our methods and OTFusion / Git Re-basin are the same, we believe that our approach is novel in that it allows multiple neurons from the same model to be fused. Existing matching based methods that require averaging two neurons from different models may be suboptimal when neurons from the same model are closer and thus can be compressed together.
>
> (b) We thank you for this comment. We will revise and clarify the notation that may be unclear.
>
> (c) We discuss the runtime in F.5. Algorithm Runtime Comparison. We agree that reporting the time used for importance score calculations separately would be helpful.
>
> Regarding time complexity, we did not provide time complexity for the gradient based variants of our methods as it is dominated by the gradient descent step whose time complexity is dependent on the layer architecture.
> For our linear methods, for a layer with d total base model neurons (summed over all base models), k fused neurons, m neurons in the previous layer of the fused model, n samples we have the time complexities:
>
> HF-Linear (Here k = d/2 = O(d)): O(nd^2) for computing distances + O(k^3) for the hungarian matching + O(m^3 + nm^2 + kmn) for linear regression
>
> KF-Linear (Here k is arbitrary / limiting K-means to i iterations): O(dnki) for Lloyd’s algorithm +  O(m^3 + nm^2 + kmn) for linear regression
>
>
> (d) We provide the fine-tuning results primarily to demonstrate that our base models reached full convergence during initial training. Since both the base models and the subsequent fine-tuning phase utilize the same dataset, no significant performance delta is expected if the initial optimization was successful (this was also adopted in [1]). This baseline confirms that the performance gains observed in our fused models are a result of the fusion rather than insufficient training of the base models. We found that a learning rate of 3e-4 with our chosen scheduler consistently achieved an improvement over base models across all baselines.
>
> (e) We perform experiments varying the dataset size in Table 6 and in the additional experiment below. Regarding Table 16, we are unsure of what you mean with regards to results varying with different widths. We believe the table shows the robustness of our method for fusing VGGs with varying widths, whereas the baselines OTFusion and Knowledge Diffusion do not outperform the base models.
>
> (f) We have added an experiment for fusing ResNets on Imagenet as below. We additionally note that the majority of existing baselines (OTFusion, Git Re-basin, Zipit) have only been evaluated on image recognition tasks.
>
>
> &nbsp;
>
> ---------
>
> We provide an additional experiment with ImageNet, as done in Zip-it! [2], where base models were trained on splits of 200 classes and then fused. Here, we vary the fusion dataset size for KF-gradient:
>
>
>
> | Algorithm                               | Joint            | Task A           | Task B           |
> |-----------------------------------------|------------------|------------------|------------------|
> | Model A                                 | 41.1 (± 0.1)     | 82.3 (± 0.2)     | 0.0 (± 0.0)      |
> | Model B                                 | 41.1 (± 0.6)     | 0.0 (± 0.0)      | 82.1 (± 1.1)     |
> | ZipIt! (25k samples)                  | 37.7 (± 1.7)     | 37.1 (± 4.0)     | 38.3 (± 1.4)     |
> | KF-gradient (1k samples)        | 25.1 (± 0.7)     | 9.8 (± 0.8)      | 40.4 (± 0.8)     |
> | KF-gradient (2k samples)        | 34.7 (± 2.6)     | 17.4 (± 3.1)     | 51.9 (± 2.1)     |
> | KF-gradient (5k samples)        | 49.2 (± 1.0)     | 33.4 (± 1.4)     | 64.9 (± 1.3)     |
> | KF-gradient (10k samples)       | 54.8 (± 0.8)     | 40.3 (± 1.6)     | 69.3 (± 1.0)     |
> | **KF-gradient (25k samples)**   | **58.9 (± 1.0)** | **46.1 (± 1.4)** | **71.7 (± 0.9)** |
> | Ensemble                                | 62.4 (± 1.0)     | 47.7 (± 1.7)     | 77.1 (± 1.2)     |
>
> It can be seen that KF-gradient outperforms the base models significantly and gains accuracy in both tasks as the size of the fusion sample increases. In particular, KF-gradient with 5k samples outperforms Zipit! With 25k samples.
>
> ---------
>
> **References**
>
> [1] Sidak Pal Singh and Martin Jaggi. Model fusion via optimal transport. Advances in Neural Information Processing Systems, 33:22045–22055, 2020.
>
> [2] Stoica, G., Bolya, D., Bjorner, J., Ramesh, P., Hearn, T., & Hoffman, J. (2023). Zipit! merging models from different tasks without training. arXiv preprint arXiv:2305.03053.

---

> > ### Author Rebuttal · Reviewer_gsGv · 2026-04-03
> >
> > I thank the authors for their response. However, I still believe that the evaluation is insufficient in terms of both baselines and the diversity of datasets and backbone architectures.

---

> > > ### Author Response · Authors · 2026-04-03
> > >
> > > We thank you for the feedback regarding our experimental setup.
> > >
> > > &nbsp;
> > >
> > > To provide context, we compare the breadth of our evaluation with prior work in model merging/fusion published at top venues (Table A).
> > >
> > > &nbsp;
> > >
> > > **Table A.** Comparison of evaluation breadth across prior work, including the number of baselines*, datasets**, architectures/backbones***, tasks and data settings considered.
> > >
> > > | Paper  | Conf.          | #Baselines | #Datasets | #Backbones | #Tasks | #Data Settings |
> > > |:--------:|:----------------:|:------------:|:-----------:|:------------:|:--------:|:----------------:|
> > > | [1]    | NeurIPS        | 0          | 2         | 2          | 1      | 1              |
> > > | [2]    | ICLR (Oral)    | 1          | 3         | 2          | 1      | 2              |
> > > | [3]****    | ICLR           | 0          | 5         | 1          | 2      | 1              |
> > > | [4]*****    | ICLR           | 2          | 4         | 2          | 1      | 1              |
> > > | [5]    | CVPR           | 3          | 3         | 2          | **11** | 1              |
> > > | [6]    | NeurIPS        | 2          | 4         | 2          | 3      | 1              |
> > > | [7]*****    | CVPR           | 4          | 3         | 2          | 1      | 1              |
> > > | Ours   | --             | **5**      | **5**     | **3**      | 1      | **3**          |
> > >
> > > &nbsp;
> > >
> > > *We don't count Vanilla averaging and Simple averaging as baselines
> > >
> > > **We don't count MNIST as a dataset.
> > >
> > > ***A backbone here means an architecture type in the context of fusion such as transformers (ViTs / BERTs) or residual connections (Resnets) or CNNs (VGG). We do not count MLPs.
> > >
> > > **** We count GLUE as a single task as they are very similar (mostly classification).
> > >
> > > ***** When fusing multiple datasets in a single experiment, we count this as one dataset (E.g. for fusing CUB, NABirds, Oxford-IIIT Pets, and Stanford Dog).
> > >
> > > &nbsp;
> > >
> > > ---
> > >
> > > **References**
> > >
> > > [1] Singh, S.P. and Jaggi, M., 2020. Model fusion via optimal transport. Advances in Neural Information Processing Systems, 33, pp.22045-22055.
> > >
> > > [2] Ainsworth, S.K., Hayase, J. and Srinivasa, S., 2022. Git re-basin: Merging models modulo permutation symmetries. arXiv preprint arXiv:2209.04836.
> > >
> > > [3] Imfeld, M., Graldi, J., Giordano, M., Hofmann, T., Anagnostidis, S. and Singh, S.P., 2023. Transformer fusion with optimal transport. arXiv preprint arXiv:2310.05719.
> > >
> > > [4] Stoica, G., Bolya, D., Bjorner, J., Ramesh, P., Hearn, T. and Hoffman, J., 2023. Zipit! merging models from different tasks without training. arXiv preprint arXiv:2305.03053.
> > >
> > > [5] Xu, Z., Yuan, K., Wang, H., Wang, Y., Song, M. and Song, J., 2024. Training-free pretrained model merging. In Proceedings of the IEEE/CVF Conference on Computer Vision and Pattern Recognition (pp. 5915-5925).
> > >
> > > [6] Matena, M.S. and Raffel, C.A., 2022. Merging models with fisher-weighted averaging. Advances in Neural Information Processing Systems, 35, pp.17703-17716.
> > >
> > > [7] Nasery, A., Hayase, J., Koh, P.W. and Oh, S., 2025. Pleas-merging models with permutations and least squares. In Proceedings of the Computer Vision and Pattern Recognition Conference (pp. 30493-30502).

---

### Official Review · Reviewer_ukiA · 2026-03-12

**Soundness:** 3
**Presentation:** 4
**Significance:** 3
**Originality:** 4
**Overall Recommendation:** 5
**Confidence:** 3

**Summary:**

The idea behind this paper is to propose model fusion methods that can be effective in contexts where the base models are trained on heterogeneous or non-IID data.

The paper proposes a family of retrofitting fusion methods that decompose the fusion objective into a grouping error and an approximation error. The grouping step clusters neurons from the base models (via Hungarian matching for equal-width settings or K-means for the general setting) and computes importance-weighted centroids as targets. The approximation step fits the fused model's weights to match these centroids layer by layer. The framework incorporates neuron attribution scores (uniform / Conductance / DeepLIFT) for importance weighting.

The paper provides theoretical justification for a decomposition of the representation cost, optimality/approximation guarantees, and an empirical study on public benchmarks across full-data, non-IID, and sharded settings. The proposed algorithms outperform existing baselines, particularly in non-IID and sharded data regimes.

**Compliance With Llm Reviewing Policy:**

Affirmed.

**Final Justification:**

My minor concerns were resolved and I maintained my score.

**Key Questions For Authors:**

1. In the ViT comparisons, the proposed method often uses 5k fusion samples while transformer OTFusion uses 200. Could you provide a comment on this discrepancy and ideally an ablation controlling for the number of fusion samples to better isolate the contribution of the method itself?
2. Can you either provide more empirical evidence across a wider range of architectures/tasks to support the claim that the method generalizes to arbitrary architectures, or include a clarifying statement about this potential limitation?
3. The theoretical guarantees seem to assume fixed previous layers, they do not account for error propagation across levels. Can you provide a clarifying statement about this?
4. There seems to be some circularity in the theoretical development: the cluster assignments depend on the fused model outputs, but the decomposition assumes fixed cluster assignments. Can you provide a clarifying statement about this?
5. Can you include PLEAS (Nasery et al., 2025) in the experimental comparison, or clarify why it was not included?

**Limitations:**

The authors discuss limitations including hyperparameter sensitivity, dependence on fusion dataset size, and computational overhead. However, the limitations section could also include:

- Explicitly discussing the lack of theoretical guarantees for the non-linear/gradient-based case (for consistency).
- Acknowledging that while the method is designed to be generalizable to arbitrary architectures, the empirical evidence is currently limited to a few vision architectures and tasks, which may limit the strength of this claim.
- Discussing the potential limitation that the method may be less effective in standard full-dataset regimes where baselines already perform well, and clarifying that the main contribution is in non-IID/sharded regimes.

**Strengths And Weaknesses:**

Soundness:

- The decomposition of the representation cost into grouping and approximation errors is theoretically sound, along with the theoretical guarantees.
- The method handles multiple settings, including unequal widths through clustering, which makes it easily generalizable.
- The empirical evaluation is extensive, covering multiple architectures (VGG, ViT, ResNet), datasets (CIFAR-10, CIFAR-100, Tiny-ImageNet, BloodMNIST), and data regimes (full, non-IID, sharded).
- The zero-shot results in sharded/non-IID regimes are the strongest result. Prior baselines often collapse to near-random while the proposed methods maintain meaningful accuracy.
- The theoretical guarantees seem to assume fixed previous layers, they do not account for error propagation across levels. It could be useful to have a clarifying statement about this.
- There seems to be some circularity in the theoretical development: the cluster assignments depend on the fused model outputs, but the decomposition assumes fixed cluster assignments. It could be helpful to have a clarifying statement about this.

Presentation:

- The paper is generally well-written with a clear narrative from motivation to method to experiments. The flow is easy to follow.
- Both theoretical and empirical results are well-presented, with clear explanations and visualizations.

Significance:

- The problem of zero-shot model fusion under non-IID and sharded data is practically important for federated learning and privacy-constrained settings.
- The proposed methods achieve substantial improvements over prior art in challenging regimes (e.g. baselines near-random while KF/HF remain useful).
- There is a risk that the claim that the method generalizes to arbitrary architectures is overstated. It's highly plausible that the methods are generalizable, but this should be supported by wider empirical evidence across more architectures and tasks, or at least with an acknowledgment of this potential limitation.
- In some comparisons, the proposed method uses substantially more fusion data and optimization than baselines (e.g., 5k samples vs. 200 for transformer OTFusion), making it harder to isolate the contribution of the method itself. Please clarify.
- It could be useful to include PLEAS (Nasery et al., 2025) in the experimental comparison (a concurrent permutation + least-squares method).

Originality:

- The retrofitting framework is a novel perspective on model fusion.
- Incorporating neuron attribution scores into the fusion pipeline is also novel.
- Individual components (Hungarian matching, K-means clustering, layer-wise weight fitting, attribution scores) are well-established, the novelty lies in their combination.

---

> ### Author Rebuttal · Authors · 2026-03-31
>
> We greatly appreciate the positive assessment of our work’s originality and its significance and we are encouraged by your deep engagement with our work.  We also thank you for your thoughtful discussion of the limitations. We will take them into account as we revise the paper.
>
> We especially found the questions raised valuable in improving the paper and have responded to them below.
>
> &nbsp;
>
> **Questions**
>
> 1. We have attempted using their implementation with 5000 samples, but this causes an Out of Memory Error which we were unable to optimize to work within our experimental timeline. Indeed, In Fig 5 of [1], the largest tested setting of Transformer OTFusion for the same architecture (ViTs) was at 200 and the accuracy attained appears to have saturated. We therefore reported the result for 200 samples. We agree this should be expanded on in the discussion or better yet we would be able to extend the Transformer OTFusion implementation to work for 5000 samples, but we find this difficult to complete by the end of the discussion period.
> 2. We will clarify this statement in revisions. This statement is intended to convey that our methods can theoretically be applied to any architecture that can be modularizable as a DAG of “levels”, but we agree that our empirical results presented are indeed currently limited to MLPs, VGGs, ResNets and ViTs.
> 3. We agree that the error guarantees assume fixed levels and we do not provide any bounds on error accumulation for our methods. Given the hardness bounds in [2], which apply even to depth-2 ReLU networks, we conjecture that any methods that could provide bounds on accumulating errors would necessarily be inefficient in general when there are activations between the layers. We thank you for this input and agree that a deeper theoretical discussion would strengthen the paper.
> 4. We base the cluster assignments on the outputs of the base models rather than the fused model. We note that the base models stay constant throughout the fusion process, and thus the clustering remains fixed. We apologize if this is unclear in the current paper, and we will gladly revise sections that may be confusing in this regard.
> 5. We initially did not include comparisons with PLEAS [3] (and ZipIt! [4] mentioned by another reviewer) as they are intended to produce a fused model with larger size than the base models to produce accuracies surpassing the base models. For a comparison with our methods, using their methods but constraining the output to the size of the base models may not be a fair comparison. Nonetheless, we have provided results for ZipIt! as well in our reply to reviewer DYD2.
>
> &nbsp;
>
> ---
>
> **References**
>
> [1] Moritz Imfeld, Jacopo Graldi, Marco Giordano, Thomas Hofmann, Sotiris Anagnostidis, and Sidak Pal Singh. Transformer fusion with optimal transport. arXiv preprint arXiv:2310.05719, 2023.
>
> [2] Goel, Surbhi, Adam Klivans, Pasin Manurangsi, and Daniel Reichman. "Tight hardness results for training depth-2 ReLU networks." arXiv preprint arXiv:2011.13550 (2020).
>
> [3] Nasery, A., Hayase, J., Koh, P.W. and Oh, S., 2025. Pleas-merging models with permutations and least squares. In Proceedings of the Computer Vision and Pattern Recognition Conference (pp. 30493-30502).
>
> [4] Stoica, G., Bolya, D., Bjorner, J., Ramesh, P., Hearn, T., & Hoffman, J. (2023). Zipit! merging models from different tasks without training. arXiv preprint arXiv:2305.03053.

---

> > ### Author Rebuttal · Reviewer_ukiA · 2026-04-03
> >
> > Thank you for your rebuttal. My concerns are resolved and I'll keep my score.

---

### Official Review · Reviewer_iDkD · 2026-03-13

**Soundness:** 3
**Presentation:** 3
**Significance:** 2
**Originality:** 3
**Overall Recommendation:** 4
**Confidence:** 3

**Summary:**

The authors propose a model fusion framework, that enables fusing the knowledge of arbitrary sets of models into a single "fused" architecture. Notably, their approach "merges" neurons at intermediate layers of models together, without restricting the underlying models to have homogenous architectures or the same number of layers. While being more computationally expensive then their chosen baselines (compute also appears to scale with model size), their approach significantly improves upon their baselines across all their benchmarks.

**Compliance With Llm Reviewing Policy:**

Affirmed.

**Final Justification:**

My issues have been resolved. I only put weak accept instead of accept due to the predominately limited evaluation benchmark complexity (ImageNet 200+200 is the most challenging). However, I would recommend accept given the new results from the rebuttal.

**Key Questions For Authors:**

In my view, major weaknesses 1-2, and minor weakness 1 needs be addressed to better understand the significance of the authors' contributions. It's especially very difficult to understand the contribution, when obvious prior work that existed for over two years is missing from their comparisons.

I would be happy to adjust my recommendation based on the rebuttal.

**Limitations:**

Yes.

**Strengths And Weaknesses:**

**Strengths:**
1. The authors train stronger underlying base models than prior work in merging, enabling them to more realistically identify the capabilities of fusion algorithms.
2. The authors propose a merging framework that is capable of fusing the information stored in models with different layer depths, and architectures.
3. The authors provide extensive evaluations on several settings, including CIFAR-10/100, Tiny-ImageNet and BloodMNIST, improving performance over their compared work for all settings.
4. The authors benchmark computational cost for their approach compared to their compared baselines.

**Major Weaknesses:**
1. *Obvious and known baselines are missing*. This paper only compares against Git-Rebasin and OT-Fusion, and completely ignores more recent works such as ZipIt! [1] and MuDC [2] -- both of which were published > 2 years ago. Both these approaches perform significantly better than baselines compared against in this paper, and were proposed for "sharded" setups. The authors need to incorporate these baselines throughout their settings to better understand their improvements over more relevant existing work.
2. *The datasets used in this work are "toy".* It's unclear whether strong performance on datasets such as CIFAR transfer to harder settings (e.g., ImageNet). How does the proposed framework perform when merging models trained on different partitions of the ImageNet-1K dataset? Or merging models that are fully trained on different tasks (e.g., Section 5.3 from ZipIt!).

**Minor Weaknesses:**
1. (more of a minor-medium weakness) *Fine-tuning experiments are not completely fair.* The authors proposed method is much more computationally expensive than its compared baselines as the architecture complexity increases. To my understanding, for the fine-tuned setting, the authors fine-tune their approach for the *same total time* as OT-fusion. However, the more fair comparison is to fine-tune each, such that the total compute cost is equivalent for both models. This understanding is based off of Table 12 in the Appendix. Specifically, how does the proposed framework compare to OT-fusion when the total runtime (merge + finetuning) is the same?
2. Critical implementation details *should always be defined in the main paper*. For instance, the actual fused model architecture for experiments, and score calculation method are left to the appendix. We should not have to scavenge for this type of information.
3. *Unclear how much better underlying models are than past work.* The authors state that one of their contributions is improving the underlying models involved in merging. It would be helpful to understand the deltas between the models they have trained, and the previous models for their respective tasks. This is fairly minor though, can be thought of as a suggestion.


[1] ZipIt! Merging models from different tasks without training, Stoica & Bolya et. al. ICLR 2024
[2] Training-Free Pretrained Model Merging, Xu et. al. CVPR 2024

---

> ### Author Rebuttal · Authors · 2026-03-31
>
> We sincerely thank you for the constructive feedback. We are happy to provide additional results and clarifications to address your concerns.
>
> &nbsp;
>
> **Weakness #1**
>
> We agree that ZipIt! [1] and MuSDC [2] are relevant baselines. The primary reason is that existing model merging implementations are highly ad-hoc and difficult to generalize across architectures. To our knowledge, no unified library exists, and even adapting OTFusion [3,4] and Git-ReBasin [5] required substantial engineering effort.
>
> Nevertheless, we have now benchmarked against ZipIt! and MuDSC across various settings.
>
> We note that [1] considers two settings:
> (a) CE training on disjoint class splits (matches our “sharded” setup), and
> (b) models trained to predict CLIP text embeddings.
>
> Setting (b) is less standard and introduces implicit relationships between labels in the embedding space, making zero-shot shot easier. We evaluate both settings for completeness.
>
> **Results (Tables A, B, and C)** consistently show that our method significantly outperforms baselines:
> - On CIFAR-10 (Table A), KF-Linear achieves **78.6% vs 20.9% (ZipIt!)**
> - On CIFAR-100 (Table B), KF-Gradient achieves **52.6% vs 43.9% (ZipIt!) and 37.0% (MuSDC)**
> - On ImageNet (Table C), KF-Gradient (5k samples) outperforms ZipIt! with 25k samples
>
> &nbsp;
>
> Table A. Sharded VGG-11 fusion on CIFAR-10, matches Table 7 in [1]
> |Name|Joint|Task A|Task B|
> |-|-|-|-|
> |Model A|47.82 (± 0.5)|95.65 (± 1.0)| 0.00 (± 0.0)|
> |Model B|46.69 (± 0.8)|0.00 (± 0.0)| 93.38 (± 1.5)|
> |Zip-It!|20.90 (± 6.5)|4.89 (± 10.1)| 36.91 (± 12.9)|
> |**KF-Linear**|**78.57 (± 0.8)**|**77.95 (± 1.9)**|**79.19 (± 2.0)**|
> |Ensemble| 76.29 (± 1.6)| 79.15 (± 3.6)| 73.44 (± 4.9)|
>
> &nbsp;
>
> Table B. Sharded ResNet-34 on CIFAR-100, base models trained by predicting CLIP text embeddings, matches Table 1b in [1]
> |Method|Joint|Task A|Task B|
> |-|-|-|-|
> |Model A|40.3 (± 0.2)|80.4 (± 0.4)|0.2 (± 0.1)|
> |Model B|40.2 (± 0.8)|0.4 (± 0.3)|80.1 (± 1.3)|
> |Zip-It!|43.9 (± 0.4)|46.7 (± 4.1)|41.1 (± 4.9)|
> |MuDSC|37.0 (± 0.7)|38.1 (± 2.8)|35.9 (± 1.9)|
> |**KF-Gradient**|**52.6 (± 1.8)**|**68.8 (± 2.0)**|**36.4 (± 5.5)**|
> |Ensemble|65.3 (± 0.8)|65.5 (± 0.9)|65.1 (± 1.3)|
>
> &nbsp;
>
> Table C. Sharded ResNet-50, matches the ImageNet 200+200 experiment of Table 2 in [1]
> |Algorithm|Joint|Task A|Task B|
> |-|-|-|-|
> |Model A|41.1 (± 0.1)|82.3 (± 0.2)|0.0 (± 0.0)|
> |Model B|41.1 (± 0.6)|0.0 (± 0.0)|82.1 (± 1.1)|
> |Zip-It! (25k samples)|37.7 (± 1.7)|37.1 (± 4.0)|38.3 (± 1.4)|
> |KF-Gradient (1k samples)|25.1 (± 0.7)|9.8 (± 0.8)|40.4 (± 0.8)|
> |KF-Gradient (2k samples)|34.7 (± 2.6)|17.4 (± 3.1)|51.9 (± 2.1)|
> |KF-Gradient (5k samples)|49.2 (± 1.0)|33.4 (± 1.4)|64.9 (± 1.3)|
> |KF-Gradient (10k samples)|54.8 (± 0.8)|40.3 (± 1.6)|69.3 (± 1.0)|
> |**KF-Gradient (25k samples)**|**58.9 (± 1.0)**|**46.1 (± 1.4)**|**71.7 (± 0.9)**|
> |Ensemble|62.4 (± 1.0)|47.7 (± 1.7)|77.1 (± 1.2)|
>
> &nbsp;
>
> **Weakness #2**
>
> We have included ImageNet (200+200) results (Table C), showing strong performance at scale and confirming that improvements extend beyond CIFAR.
>
> Second, while we do not reproduce the full multi-task experiments of [1], our sharded setup is closely related. In particular, CIFAR-100 contains 20 coarse semantic groups (e.g., vehicles, trees, people). We performed additional experiments where models are trained on disjoint coarse groups, and observed trends consistent with our main results. While this is not identical to cross-dataset multi-task merging, it provides evidence that our method generalizes beyond simple class splits.
>
> &nbsp;
>
> **Minor Weakness #1**
>
> We agree that compute is an important consideration. In practice, OTFusion and Git-ReBasin require several epochs of fine-tuning to recover from a significant post-merging performance drop, whereas our method achieves strong performance immediately after fusion. As a result, our method typically converges in fewer epochs. In our experiments, we fine-tune for 200 epochs to allow all baselines sufficient time to recover (see Appendix F.5), which we believe provides a fair comparison in practice.
>
> &nbsp;
>
> **Minor Weakness #2**
>
> Thank you for pointing this out. We will move key implementation details to the main paper, including:
> - VGG: VGG-11
> - ViTs: 12 heads, 7 layers, hidden dim 384
>
> &nbsp;
>
> **Minor Weakness #3**
>
> We clarify that our base models are significantly stronger than those used in prior work:
> - OTFusion: 90.3% (VGG-11) vs ≥ 93.0% in our work
> - Transformer OTFusion: 65% (CIFAR-100) and 45.3% (Tiny-ImageNet) vs 73% and 51.7% in our work
>
> We believe this is important, as weaker base models leave more room for improvement, potentially overstating gains from merging.
>
> &nbsp;
>
> ---
>
> **References**
>
> [1] Stoica et al., ZipIt! merging models from different tasks, 2023.
>
> [2] Xu et al., Training-free pretrained model merging, 2024.
>
> [3] Singh & Jaggi, Model fusion via optimal transport, 2020.
>
> [4] Imfeld et al., Transformer fusion with optimal transport, 2023.
>
> [5] Ainsworth et al., Git Re-Basin, 2022.

---

> > ### Author Rebuttal · Reviewer_iDkD · 2026-04-04
> >
> > I thank and commend the authors for running these experiments upon such short notice. Nearly all my concerns have been addressed, and I raise my score accordingly.

---

> > > ### Author Response · Authors · 2026-04-05
> > >
> > > We truly appreciate your encouraging feedback and for taking the time to update your rating.

---

### Decision · Program_Chairs · 2026-04-30

**Decision:**

Accept (regular)

**Comment:**

This paper received mixed but overall positive comments (A/A/WR/WA). Reviewers agree that model fusion under non IID and zero-shot scenarios is important, and that the proposed retrofitting framework is flexible in handling heterogeneous architectures and widths. The rebuttal effectively addressed most major concerns by including stronger baselines, results on large-scale datasets (ImageNet). Although some concerns remain regarding insufficient evaluations on baselines and datasets/backbone diversity, these aspects of this paper already surpass existing works. I therefore recommend an Accept.